# The combination of CHK1 inhibitor with G-CSF overrides cytarabine resistance in human acute myeloid leukemia

Alessandro Di Tullio[1], Kevin Rouault-Pierre [1,3], Ander Abarrategi[1], Syed Mian[1,2], William Grey[1], John Gribben[3], Aengus Stewart[4], Elizabeth Blackwood[5] & Dominique Bonnet [1]

Cytarabine (AraC) represents the most effective single agent treatment for AML. Nevertheless, overriding AraC resistance in AML remains an unmet medical need. Here we show that the CHK1 inhibitor (CHK1i) GDC-0575 enhances AraC-mediated killing of AML cells both in vitro and in vivo, thus abrogating any potential chemoresistance mechanisms involving DNA repair. Importantly, this combination of drugs does not affect normal long-term hematopoietic stem/progenitors. Moreover, the addition of CHK1i to AraC does not generate de novo mutations and in patients' samples where AraC is mutagenic, addition of CHK1i appears to eliminate the generation of mutant clones. Finally, we observe that persistent residual leukemic cells are quiescent and can become responsive to the treatment when forced into cycle via granulocyte colony-stimulating factor (G-CSF) administration. This drug combination (AraC+CHK1i+G-CSF) will open the doors for a more efficient treatment of AML in the clinic.

[1] Hematopoietic Stem Cell Laboratory, The Francis Crick Institute, 1 Midland Road, NW1 1AT London, UK. [2] King's College London School of Medicine, Department of Haematological Medicine, The Rayne Institute, 123 Coldharbour Lane, SE5 9NU London, UK. [3] Department of Haemato-Oncology, Barts Cancer Institute, Queen Mary University of London, Chaterhouse Square, EC1M 6BQ London, UK. [4] Bioinformatic Core, The Francis Crick Institute, 1 Midland Road, NW1 1AT London, UK. [5] Genentech, 1 DNA Way, South San Francisco, CA CA94 080, USA. Correspondence and requests for materials should be addressed to D.B. (email: dominique.bonnet@crick.ac.uk)

Despite increased understanding of the pathogenesis and biology of AML, patient outcomes remain poor, with a median survival of ~1 year with standard treatment. At present, Cytarabine (AraC) remains the single most effective cytotoxic agent available for the treatment of AML, which provides high early remission rate, especially in younger patients[1]. However, most AML patients relapse and succumb to their disease[1, 2]. Furthermore, elderly patients (>60 years old) cannot tolerate high-dose therapy, and current treatment protocols show an especially poor drug response in these patients. Thus, identifying strategies for overcoming AraC resistance and enhancing its efficacy remains as a high unmet clinical need.

AraC is a DNA nucleoside analog, which exerts its cytotoxic effects by disrupting normal DNA synthesis through direct incorporation in extending DNA strands. As yet, a few mechanisms of AraC resistance have been described, including deregulation of AraC metabolism[3–5] and cell quiescence, but one of the most critical mechanisms of resistance is through the DNA damage response (DDR)[6–9]. This is an adaptable response to genomic DNA damage, which includes cell cycle checkpoints, DNA repair, and apoptosis[10]. Cell cycle arrest, via checkpoint activation, is a crucial task that permits repair of DNA lesions and triggers apoptosis when damage is irreparable, thereby preserving genomic integrity. This function is largely carried out by one critical component of the DDR, the Checkpoint kinase 1 (CHK1), which is activated when replication inhibitors, including cytarabine[6–9, 11], cause DNA polymerases to stall but allow DNA helicases to advance, creating regions of single-stranded DNA. The latter causes the activation of the ataxia telangiectasia mutated and Rad3-related (ATR) kinase[12, 13], which stimulates a phosphorylation cascade, activating CHK1, which in turn phosphorylates CDC25A, leading to CDC25A degradation and cell cycle arrest[14].

In AML, a recent report even suggested that expression and/or protein level of CHK1 was associated with poor risk outcome[15]. Thus, over the past decade, literature has expanded in support of

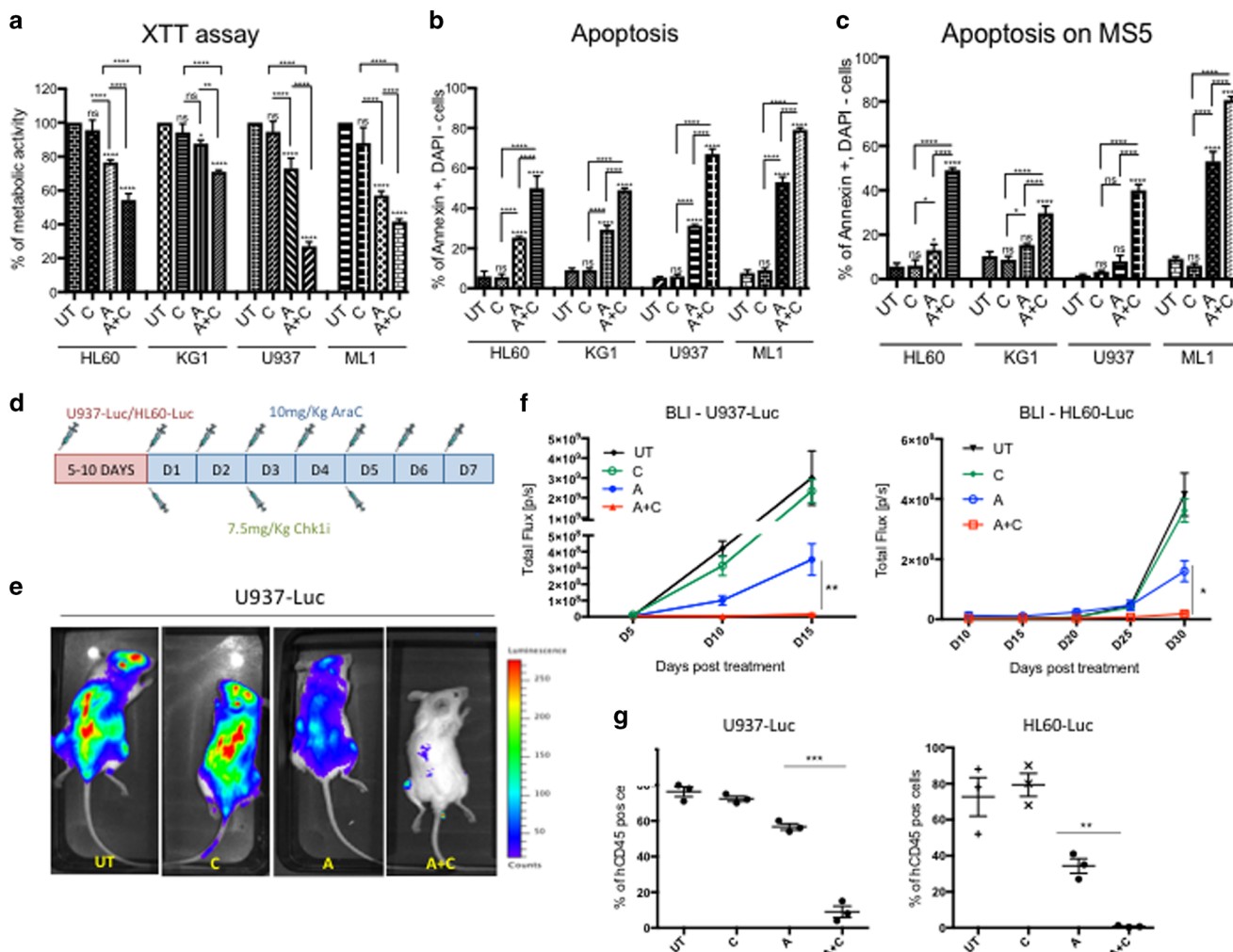

**Fig. 1** CHK1i enhances the cytotoxicity of AraC in different AML cell lines in vitro and in vivo. **a** Percentage of cell metabolic activity in four AML cell lines after 1-day treatment with AraC (A) and AraC+CHK1i (A+C) normalized on the untreated (UT) control. (UT $n = 3$; C $n = 3$; A $n = 3$; A+C $n = 3$). **b** Percentage of apoptosis with the same cell lines and treatment conditions as in **a**. (UT $n = 3$; C $n = 3$; A $n = 3$; A+C $n = 3$). **c** Percentage of apoptotic cells as in **b**, but the cell lines were seeded on an irradiated MS5 stromal layer. (UT $n = 3$; C $n = 3$; A $n = 3$; A+C $n = 3$). **d** Treatment regimen for AraC and CHK1i with U937-Luc and HL60-Luc, in vivo. **e** In vivo bioluminescence imaging (BLI) 15 days after Luc-marked cell injection under different treatment conditions (upper panel U937; lower panel, HL60). **f** Kinetics of tumor proliferation by BLI signal quantification, up to 15 days after U937-Luc injection (left panel) and 30 days after HL60-Luc injection (right panel). n = 3 animals per cohort. **g** Percentage of U937-Luc cells in the bone marrow of the transplanted mice 3 days post treatment (D15) (left panel) and HL60-Luc (D30) (right panel). Each dot represents an individual mouse. (UT $n = 3$; A $n = 3$; A+C $n = 3$). *$p < 0.05$, **$p < 0.001$, ***$p < 0.001$, ****$p < 0.0001$

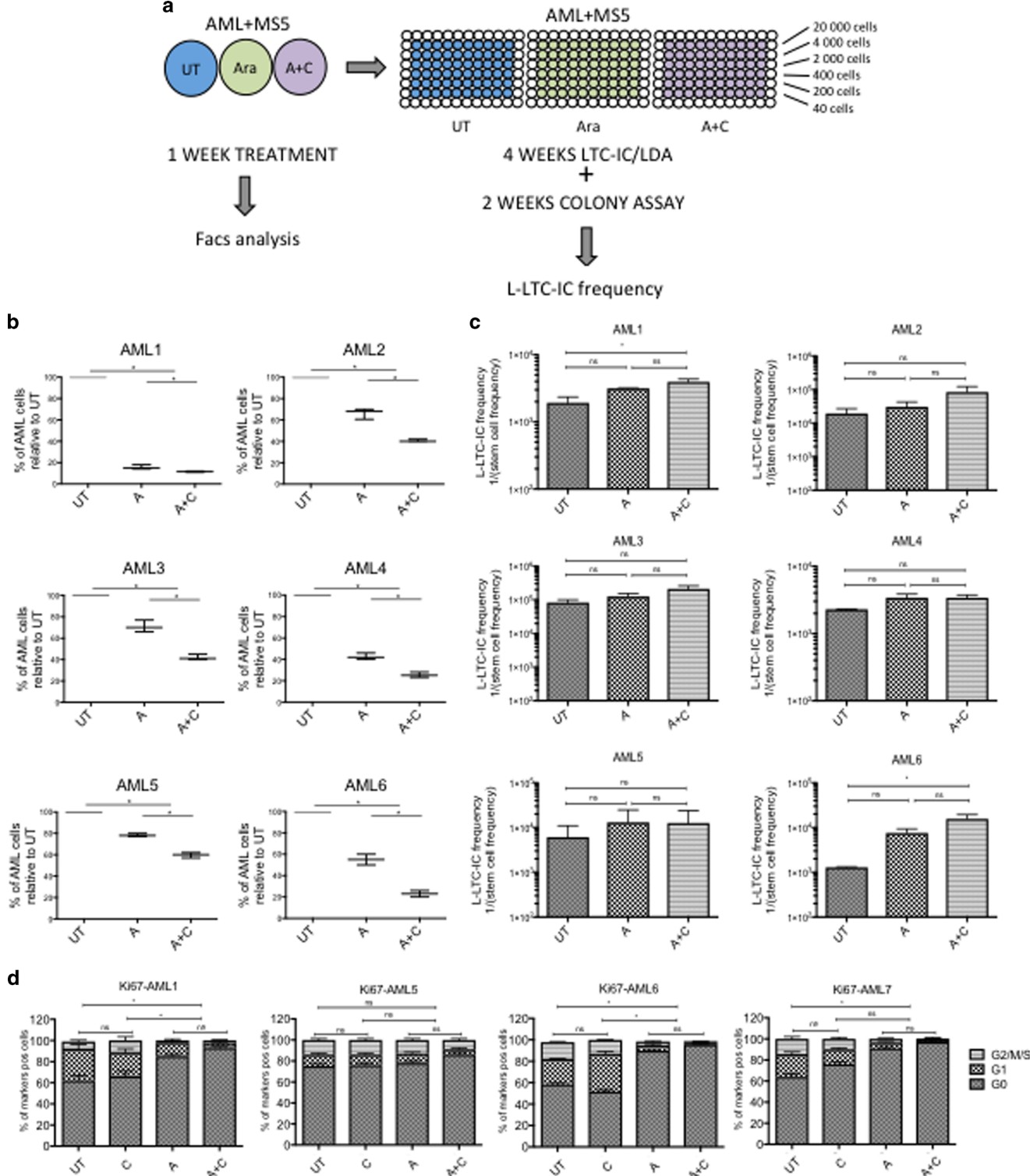

**Fig. 2** CHK1i enhances AraC efficacy to kill primary AML samples in vitro, but does not increase its effect on L-LTC-ICs depletion. **a** Scheme of AML short-term culture (STC) and long-term culture (LTC) analysis, using different treatment conditions. Three independent LTC-IC/LDA assays were performed in duplicate each time. **b** Percentage of AML survival after treatment, using counting beads, and normalizing to the untreated condition. (UT $n = 3$; A $n = 3$; A+C $n = 3$; NS = not significant). **c** L-LTC-IC frequency of AML subjected to 1 week of treatment, 4 weeks of LTC in a limiting dilution assay (LTC-LDA) and 2 weeks in methylcellulose. (UT $n = 3$; A $n = 3$; A+C $n = 3$; NS = not significant). **d** Cell cycle analysis of four AML samples 1 week after treatment, looking at Ki67/DAPI expression. (UT $n = 3$; C $n = 3$; A $n = 3$; A+C $n = 3$). Statistical test here related to cells in G0. *$p < 0.05$

targeting CHK1 to shorten the opportunity for lesion repair and thereby enhance the effect of genotoxic agents, including ionizing radiation, alkylating agents, nucleoside analogs, and topoisomerase inhibitors. Thus, inhibition of CHK1 function, using either gene depletion or selective small chemical inhibitors, has been shown to potentiate cell killing in a wide range of tumors. In many of these studies, although not all, the degree of sensitization was greater in tumor cells deficient for TP53 than in their wild-type counterparts[9, 16–20]. Recent reports further suggest that pharmacological suppression of CHK1 enhances the efficacy of conventional genotoxic agents, most importantly AraC, lending support for the deployment of CHK1 inhibitors (CHK1i) in combination therapy[21–24].

Here we use GDC-0575, an orally bioavailable CHK1 inhibitor with an IC50 of 1.2 nM. We show that this CHK1 inhibitor in combination with AraC enhances the killing of primary AML cells ex vivo by inducing apoptosis. Using an in vivo xenograft model, we confirm the enhancing effect of CHK1 inhibition with AraC using both AML cell lines and human primary AML cells. Remarkably, we demonstrate that CHK1i+AraC therapy does not affect the normal long-term HSPC compartment. Moreover, it has been recently described that AraC treatment can also induce the formation of de novo mutations in AML patients[25]. Here we provide evidence in our in vivo model that AraC+CHK1i therapy does not create new mutations, and that CHK1i prevents the survival of AraC-induced mutagenic clones. Finally, we observe that the persistent residual leukemic cells are quiescent and can become sensitive to the combination treatment by inducing their entry into the cell cycle via the use of granulocyte colony-stimulating factor (G-CSF). Our data provide a safe way to override concertedly both DNA repair- and cell quiescence-mediated resistance to AraC therapy in AML.

## Results

**CHK1i GDC-0575 structure and selectivity**. The novel CHK1 inhibitor GDC-0575 (see structure, Supplementary Fig. 1) has been tested for its selectivity against CHK1 and more than 450 kinases and disease-relevant mutant variants using the KINO-MEscan assay (see kinase selectivity profile, Supplementary Table 1).

**CHK1i GDC-0575 enhances the cytotoxicity of AraC in vitro**. Although CHK1i have been already described in vitro to sensitize AML cell lines to DNA-damaging agents[18, 20–22, 24], the identification of selective inhibitors that may decrease the side effects, and be efficacious against the large heterogeneity present in AML patients, remains a big challenge.

We tested here the effect of GDC-0575 in combination with AraC on a few established AML cell lines. First, we showed that AraC (A) at 500 nM for 24 h induced cellular DNA damage based on $\gamma$H2Ax positivity using either immunofluorescence (IF) or FACS analysis (Supplementary Fig. 2A, B respectively), which in turn activated CHK1 (Ser345 phosphorylation) (Supplementary Fig. 2A). We confirmed the activation of CHK1 mediated by AraC treatment by western blot (WB) showing an increase in the auto-phosphorylation of CHK1 at Ser296 (Supplementary Fig. 2C) and inhibition of CDC25A phosphatase activity (increase in CDC25A inactive phosphorylated form) leading to an accumulation of the catalytically inactive Tyr[15]-phosphorylated CDK2 (Supplementary Fig. 2D). We next validated that GDC-0575 at 100 nM blocked the activation of CHK1 induced by AraC via decrease in the level of Tyr[15]-phosphorylated CDK2 (Supplementary Fig. 2D). Next, we tested CHK1i (C) toxicity by treating the AML cells with 100 nM GDC-0575 for 24 h and found that it had no impact on viability compared to the untreated control

(UT), as illustrated by the XTT metabolic assay (Fig. 1a). When combining AraC with CHK1i (A+C), we observed a significant reduction in cell viability (Fig. 1a) and an increase in apoptosis (Fig. 1b) in all AML cell lines. Interestingly, the effect of CHK1i was even more pronounced when culturing the AML cells on top of the mesenchymal stromal layer MS5, which is known to reduce AraC-induced AML apoptosis[26] (Fig. 1c). We confirmed this data using a lower dose of AraC (100 nM) (Supplementary Fig. 3A), a short hairpin RNA against Chk1 (KD Chk1) (Supplementary Fig. 3B) and an inhibitor against ATR (ATRi), which is upstream of CHK1 (Supplementary Fig. 3C). As expected, ATRi plus CHK1i had no additive effect on AML-induced apoptosis.

**CHK1i increases the effect of AraC on AML cell lines in vivo**. In order to investigate whether the effect of A+C observed in vitro in AML cell lines could be confirmed in vivo, we generated two stable luciferase-expressing AML cell lines (HL60-Luc and U937-Luc) and monitored their viability post treatment by biolumi-nescence imaging (BLI), a previously described non-invasive technique for in vivo studies[27]. We injected 2 million U937-Luc or HL60-Luc cells into NOD/Scid gamma IL2Rγnull (NSG) mice and followed the treatment regimen as illustrated in Fig. 1d. We monitored the engraftment kinetics and initiated treatment of the mice when they reached 1% engraftment. Strikingly, at day 15 (U937-Luc) or 30 (HL60-Luc) post injection, a significant decrease in the expansion of AML was observed when mice received AraC alone, but an almost complete eradication of the leukemic burden was detected when co-treating with AraC and CHK1i, as revealed by BLI for mice transplanted with U937-Luc cells (Fig. 1e). The quantification of the luminescent signal is illustrated in Fig. 1f (left panel, U937-Luc; right panel, HL60-Luc) and confirmed by FACS analysis on the BM cells at the time of killing (Fig. 1g; left panel, U937-Luc; right panel, HL60-Luc).

All together, these results show that CHK1 inhibition enhances the efficacy of AraC for the treatment of AML cell lines in vivo.

**CHK1i increases AraC response on primary AML cells in vitro**. Next, we investigated whether the combination of A+C was beneficial not only in AML cell lines, but also in primary AML in vitro. We tested six samples representing different cytogenetic risk groups (Supplementary Table 2). We cultured them on the murine stromal cell line MS5 in hypoxic conditions, as previously described[28] and treated them for 1 week with 500 nM AraC and/or 100 nM Chk1 (Fig. 2a). We saw a significant decrease in AML survival when using the co-treatment (Fig. 2b). Specifically, we noticed a stronger clearance of the growing cells on top of the stroma after A and A+C treatment, leaving the quiescent cell cluster underneath it unaffected. We confirmed this by showing a significant increase in the percentage of quiescent cells (G0) by Ki67/DAPI staining after both A and A+C treatment (Fig. 2d). In order to investigate whether the co-treatment was also affecting the leukemic long-term culture-initiating cells (L-LTC-IC) and not only the leukemic blasts, we collected the surviving cells after 1 week of treatment and seeded them at limiting dilution in a long-term culture assay, as previously described[28]. We kept the cells on irradiated MS5 changing the medium every week and after 4 weeks, cells were seeded on methylcellulose for an extra 2 weeks. At the end of the assay, we counted the colonies and calculated the L-LTC-IC frequency using an online web tool (http://bioinf.wehi.edu.au/software/elda/index.html). In general, L-LTC-IC frequency remained similar or decreased with A and/or A+C treatment compared to untreated controls, indicating that the residual non-affected quiescent cells were not enriched in L-LTC-ICs (Fig. 2c). For AML patients 1, 4, and 6, a significant proportion of L-LTC-IC was actually sensitive to AraC treatment

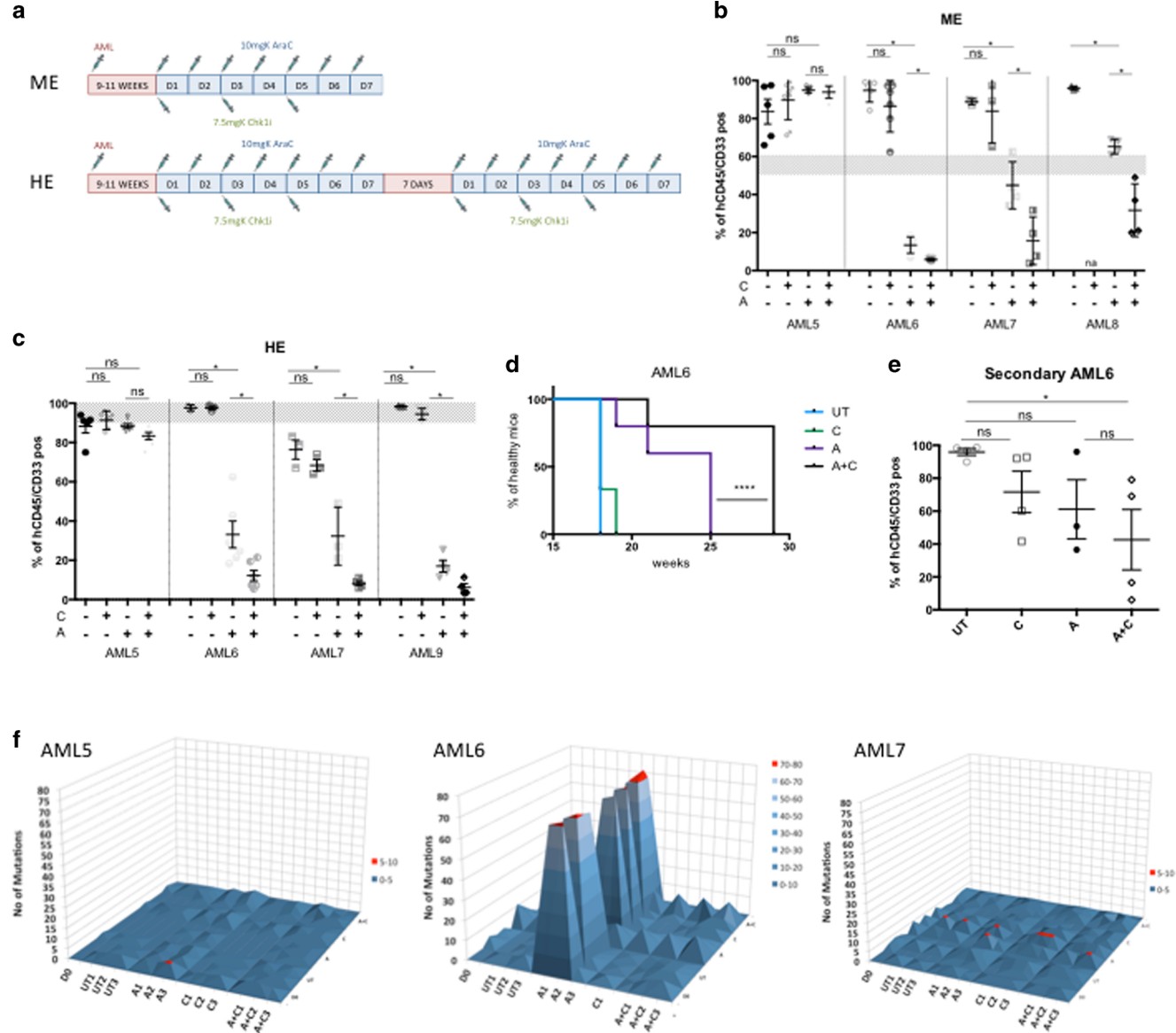

**Fig. 3** CHK1i enhances the cytotoxicity of AraC in different primary AML models in vivo and reduces mutagenic potential. **a** Scheme of treatment regimen for AML patient samples starting with medium engraftment (ME) and high engraftment (HE). **b** Percentage of human cells in the bone marrow of mice transplanted with four primary AML patient samples. The treatment started at ME (gray area) and the analysis was performed on BMs of killed mice 1 week post treatment. Each dot represents an individual mouse. (AML5 UT $n = 5$; C $n = 5$; A $n = 5$; A+C $n = 6$; AML6 UT $n = 5$; C $n = 6$; A $n = 5$; A+C $n = 5$; AML7 UT $n = 3$; C $n = 3$; A $n = 4$; A+C $n = 4$; AML8 UT $n = 3$; C $n = 0$; A $n = 4$; A+C $n = 4$; total number of mice = 67). **c** Same as in **b** but the treatment started at HE (gray area). (AML5 UT $n = 5$; C $n = 3$; A $n = 5$; A+C $n = 6$; AML6 UT $n = 3$; C $n = 3$; A $n = 6$; A+C $n = 6$; AML7 UT $n = 3$; C $n = 3$; A $n = 3$; A+C $n = 4$; AML9 UT $n = 3$; C $n = 5$; A $n = 5$; A+C $n = 4$; total number of mice = 67). **d** Sickness curves monitoring AML6-injected mouse under different treatment conditions at various time points. Mice were euthanized when a 20% reduction of body weight was reached (UT $n = 3$; C $n = 3$; A $n = 4$; A+C $n = 4$). We used the Gahan–Breslow–Wilcoxon for statistical analysis. **e** Secondary transplantation of hCD45/CD33[+] cells, pooled from AML6-injected mice 1 week post treatment. Each dot represents an individual mouse (UT $n = 4$; C $n = 4$; A $n = 3$; A+C $n = 4$). **f** 3D plots showing the changes in the mutation spectrum for each patient. Mutations present at D0 sample were compared to UT, A, C and A+C mice. *$p < 0.05$; ****$p < 0.00005$ and ns: non-significant

alone and the addition of CHK1i did not affect the L-LTC-IC frequency (Fig. 2c). For the other three AML samples (2, 3, and 5), no significant decrease in frequency in L-LTC-IC was observed after A±C treatment.

In summary, the addition of CHK1i to AraC treatment enhances the clearance of proliferating primary AML cells as a bulk population in vitro, but it does not significantly increase L-LTC-ICs elimination, over AraC treatment alone.

**CHK1i plus AraC treatment delays AML relapse in vivo.** We next tested the effect of A+C combination on primary AML

samples in vivo using the NSG xenograft mouse model. We studied the effect of chemotherapy at medium (ME = 50–60%) and high engraftment (HE = ~90–100%) levels, which would resemble different stages of leukemia progression in humans. The treatment regimen differed for the two groups: ME mice were treated for 1 week, whereas HE mice received two courses of 1-week treatment separated by 1-week rest, according to the treatment regime depicted (Fig. 3a). Patient samples from different cytogenetic risk groups were tested (Supplementary Table 2). Bone marrow aspirates of mice transplanted with human AML cells were analyzed to monitor their engraftment

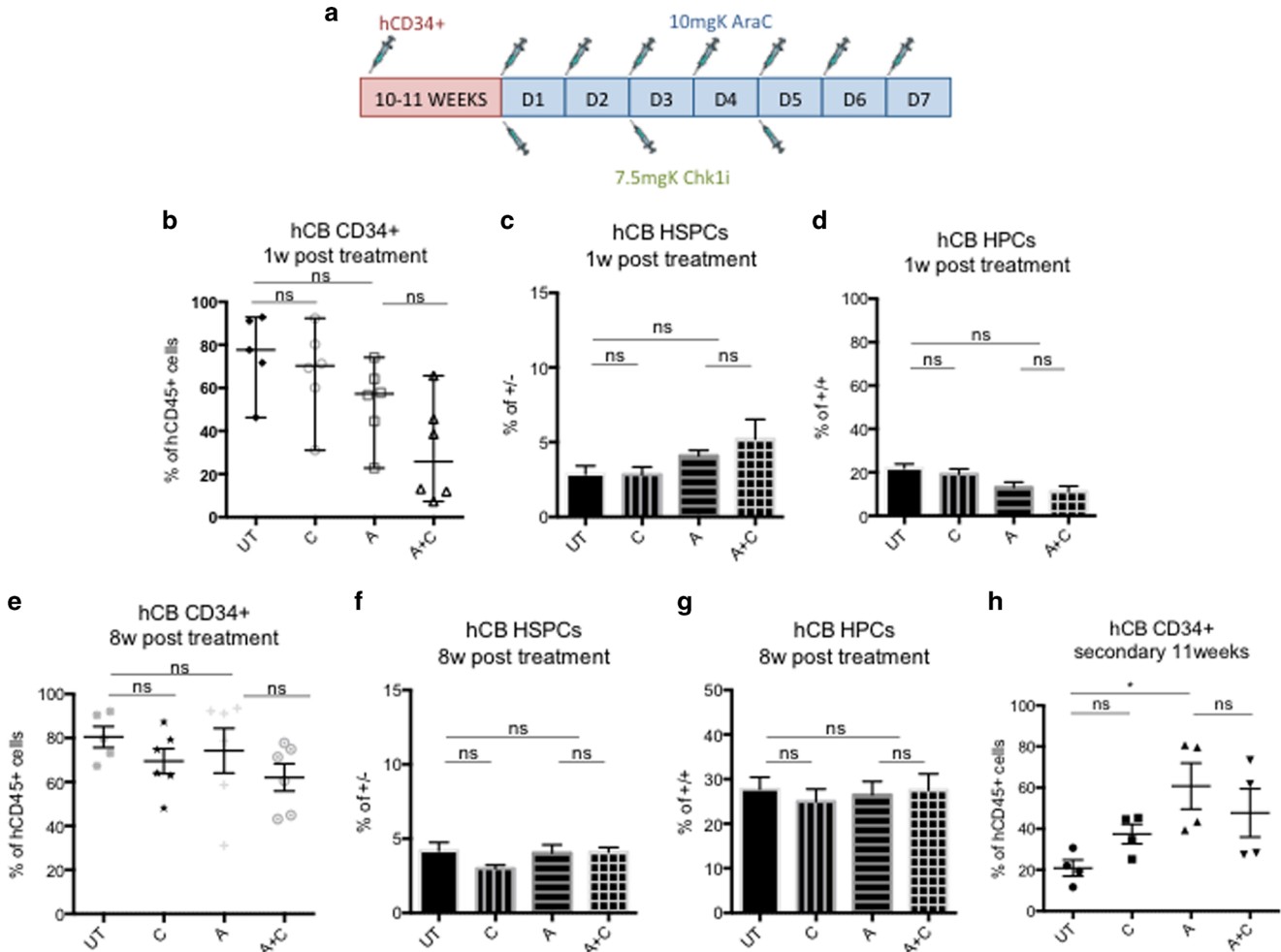

**Fig. 4** CHK1i plus AraC treatment does not affect human hematopoiesis derived from cord blood in vivo. **a** Treatment regimen for mice transplanted with hCB CD34$^+$ cells. **b** Percentage of human cells in the BM aspirates of hCB-transplanted mice 1 week post treatment. Each dot represents an individual mouse (UT $n = 5$; C $n = 6$; A $n = 6$; A+C $n = 6$. ns = not significant. Total number of mice = 23). **c** Percentage of HSPCs (Lin$^-$CD34$^+$CD38$^-$) in the BM aspirates of hCB-transplanted mice 1 week post treatment. The analysis was performed on the same mice as in **b**. **d** Same as in **c**, but percentage of HPCs (Lin$^-$CD34$^+$CD38$^+$). **e** Same as in **b** but at 8 weeks post treatment when mice were killed for analysis. **f** Same as in **c** but 8 weeks post treatment. **g** Same as in **d** but 8 weeks post treatment. **h** Secondary transplantation of hCD45$^+$ cells, pooled from hCB-injected mice 1 week post treatment. Each dot represents an individual mouse (UT $n = 4$; C $n = 4$; A $n = 4$; A+C $n = 4$). *$p < 0.05$ and ns = non-significant

level (9–11 weeks after injection) and their viability after treatment (1 week post treatment). After treatment at ME level, we observed in UT and C-treated mice, an increase of the AML burden compared to pre-treatment level (gray zone in Fig. 3b), and a significant decrease in A-treated mice (Fig. 3b) for three out of the four patients tested. The addition of CHK1i to AraC treatment in these three samples significantly decreased leukemic engraftment even further. This was not the case for patient AML5, which appeared to be insensitive to AraC. Similar results were obtained when starting the treatment when engraftment level had reached a higher baseline level (HE) (Fig. 3c). A remarkable example of AML reduction when using the co-treatment could be observed by FACS analysis, with a decrease of human engraftment in mouse BM from ~90% before treatment to 3.5% post treatment (see Supplementary Fig. 4A, patient AML9). Here the analysis of the spleens clearly showed a drastic reduction in size specifically in the co-treatment group (Supplementary Fig. 4C). Moreover, we analyzed the AML engraftment levels in the BMs also by immunohistochemistry (CD45$^+$ IHC) and checked the cell cycle status of the resistant cells by Ki67 staining (Supplementary Fig. 4B, AML9). It was clear that the few remaining human cells, after the co-treatment, were quiescent.

Similar results were obtained with patient AML6 (Supplementary Fig. 4E). In addition to the reduction in disease burden, we detect a strong delay in leukemic relapse, as suggested by the percentage of healthy mice over time (Fig. 3d, AML6 and Supplementary Fig. 4D, AML9).

The next question we addressed was whether residual surviving cells were enriched in LICs. To determine this, we performed secondary transplantation assays, retrieving cells from AML6-transplanted primary mice, pooling them by condition and injecting the same number of cells into secondary mice. There was a significant reduction in the level of human engraftment in A+C-treated mice, but not A or C-treated alone (Fig. 3e), indicating that in vivo the single treatment was sparing quiescent AML cells, which contain LICs activity.

Finally, it has been reported that AraC treatment can induce de novo mutations in AML patients[25]. To test whether the addition of CHK1i could have a similar effect, we performed whole-exome sequencing on three primary AML patient samples (Supplementary Table 3) before and after transplantation in mice and 1-week after the last treatment (Supplementary Table 4). We saw that adding CHK1i to AraC treatment did not generate new mutations compared to Day 0 and/or to the UT samples (Fig. 3f and

Supplementary Fig. 5). Importantly, we noticed that in one patient (AML6) where AraC had mutagenic effects (Fig. 3f and Supplementary Fig. 5), the addition of CHK1i prevented the survival of these new mutated clones, thus averting the acquisition of new driving mutations, a significant contributor to relapse in AML.

Taken together, these data suggest that CHK1 inhibition enhances the killing effect of AraC on primary AML cells in vivo. The double treatment is not mutagenic and might also prevent the survival of AraC-derived mutated clones, reducing the likelihood of relapse.

**CHK1i plus AraC treatment does not affect normal HSPCs**. It is fundamental for new AML treatments to evaluate the effect of the drug combination on normal human hematopoiesis. To investigate this, we sorted $CD34^+$ cells from human cord blood (hCB) and injected them into NSG mice. When they reached ~70% human engraftment level, we divided the mice into four groups and then followed the treatment regimen illustrated in Fig. 4a. One week post treatment, a small but insignificant decrease could be seen in human engraftment for all treatment types compared to UT controls (Fig. 4b). We investigated this small decrease further to analyze which hematopoietic sub-population was most affected by the treatment. Hematopoietic stem and progenitor cells (HSPCs, $Lin^-CD34^+CD38^-$) were importantly not affected by the double treatment (Fig. 4c), whereas hematopoietic precursor cells (HPCs, $Lin^-CD34^+CD38^+$) showed a slight reduction in cell viability (Fig. 4d), even though the difference between A+C and A-treated groups was not significant. Furthermore, we investigated whether the treatment protocol could affect the long-term outlook of human hematopoiesis and therefore we killed the mice 8 weeks post treatment and analyzed the human engraftment in the BMs. We found a high percentage of human hematopoietic cells in all treatment groups (Fig. 4e). Analysis of the HSPC and HPC compartments showed that the percentage of HSPCs (Fig. 4f) and HPCs (Fig. 4g) was not affected by the treatment. Furthermore, we studied the functionality of the HSPCs after chemotherapy by performing secondary transplantations. Eleven weeks post transplantation, we checked the engraftment levels of secondary mice and actually observed an increase in human cells in mice injected with A and A+C group compared to UT and C groups (Fig. 4h). This result suggests that the treatment of human hematopoietic cells in primary mice selected for a quiescent population, which was enriched in HSCs. Finally, considering that hCB HSCs present different characteristics when compared to their human adult bone marrow (hBM) counterparts, most importantly higher proliferation rate[29, 30], we decided to test the effect of the co-treatment in mice engrafted with the more quiescent adult hBM HSPCs. Following the previously described treatment protocol (Fig. 4a), we saw no significant decrease in the level of engraftment in treatment groups 1 week post treatment compared to UT controls (Supplementary Fig. 6A). Similar to hCB results, we did not detect a significant difference when comparing A-treated mice to the A+C group, either at the HSPC level (Supplementary Fig. 6B) or at the HPC level (Supplementary Fig. 6C). To investigate whether the chemotherapy could have affected the long-term outlook of human hematopoiesis, we euthanized the mice 8 weeks post treatment. Once again, we found no significant difference among the treatment groups (Supplementary Fig. 6D), at the HSPC (Supplementary Fig. 6E) or HPCs level (Supplementary Fig. 6F).

In summary, addition of CHK1i to AraC treatment does not impede normal long-term hematopoiesis derived from either hCB or hBM cells.

**G-CSF enhances the efficacy of AraC+CHK1i on AML in vivo**. The addition of Chk1i considerably improves the primary effect of AraC treatment for AML, but the remaining small fraction of quiescent cells containing LICs must be eradicated to avoid the possibility of relapse.

To test if these residual LICs survived because of their quiescent state and not because of any other resistance mechanisms, we decided to force these cells to cycle via the administration of granulocyte colony-stimulating factor (G-CSF), a well-known hematopoietic stem cell mobilizing agent[31] as previously described[32]. We thus established its mobilizing effect on AML by treating the engrafted mice according to the scheme in Supplementary Fig. 7A and analyzing BMs and blood 1 day post treatment. In AML-engrafted mice, we observed a significant increase of human cells in the blood of G-CSF-treated mice compared to UT but not in the BM (Supplementary Fig. 7B). Moreover, we confirmed that G-CSF was increasing the number of proliferating AML cells and reducing the quiescent ones in the BM (Supplementary Fig. 7C).

With these positive results, we introduced G-CSF to our treatment regimen in combination with A+C treatment (A+C+G) (Fig. 5a). Using this new treatment protocol, we tested four primary AML patient samples, starting the treatment at HE, and analyzed the human engraftment levels 1 week post treatment, as previously described. Remarkably, we noticed that all AML-engrafted mice were subject to a stronger leukemia reduction when adding G-CSF to A+C treatment (Fig. 5b), including the non-responder AML5 sample. AML7-engrafted mice offered a remarkable example of nearly complete leukemic eradication after A+C+G treatment, as shown by FACS (Fig. 5c). We noticed, by IHC, that the small amounts of spared cells after the triple treatment were still quiescent (Fig. 5d), suggesting that a longer exposure to G-CSF might be needed. To assess the functionality of those remaining cells, we performed either L-LTC-IC assay or secondary mouse transplantation from the A+C- and A+C+G-treated mice. We showed by the L-LTC-IC assay, a significant reduction in the stem cell frequency of A+C+G compared to A+C from AML7 (Fig. 5e). For AML8, where we could perform secondary transplantation, we detected around 40% engraftment in the A+C group 12 weeks post injection, whereas no detectable engraftment was seen in the A+C+G group (<0.01% detection limit) (Fig. 5f).

All together, these data show that the addition of G-CSF to the AraC+CHK1i combination therapy facilitates the elimination of the quiescent LICs. These results also suggest that residual LICs are not intrinsically resistant to AraC+CHK1i treatment, as when the LICs are forced to cycle, they become sensitive and are eradicated.

**AraC+CHK1i+G-CSF treatment does not affect normal HSPCs**. Lastly, we investigated whether the addition of G-CSF to A+C would impede normal human hematopoieisis. We first confirmed the mobilizing effect of G-CSF on both hCB and hBM $CD34^+$ cells following the treatment regimen in Supplementary Fig. 7A by analyzing mouse BM and blood 1 day post treatment. Although we did not observe a difference in terms of total number of hematopoietic cells (hCB $CD45^+$) in G-CSF-treated mice compared to UT (Supplementary Fig. 7D), we found that hCB HSPCs were mobilizing from the BM to the blood (Supplementary Fig. 7E). Similar results were found when we transplanted the mice with the "less proliferative" hBM cells. Again, we could observe a general mobilization of hBM $CD45^+$ cells into the blood (Supplementary Fig. 7F) and a clear shift of hBM HSPCs from the bone marrow into the blood (Supplementary Fig. 7G). Hence, we introduced G-CSF to our treatment regimen in combination with A+C treatment (A+C+G) (Fig. 5a). We

transplanted hBM CD34+ cells and started the treatment when mice achieved ~70% human engraftment level. Upon A+C+G treatment, we detected a significant reduction of human cells compared to A+C group, 1 week post treatment, but cells recovered over a longer period of time (8 weeks post treatment)

(Fig. 5g). The functionality of the remaining human CD34+ cells was tested by serial transplantation. We observed that cells exposed to the triple treatment were able to engraft in secondary mice at the same frequency as A+C and UT group (Supplementary Table 5).

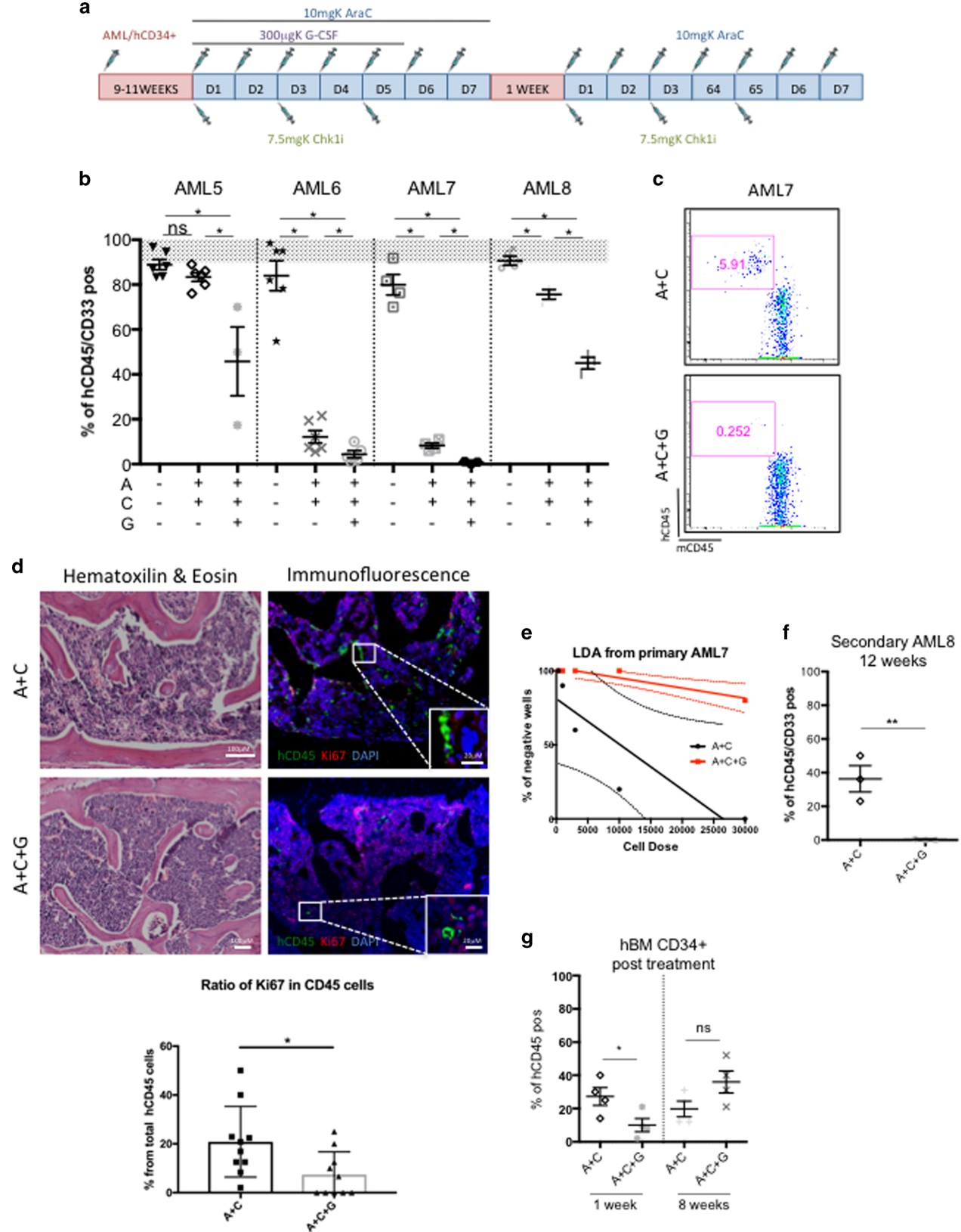

These results suggest that even though G-CSF exhibits a mobilizing effect on normal HSPCs, the triple therapy regimen used in this study does not impede normal long-term hematopoiesis.

## Discussion

DNA damage checkpoints have recently drawn great interest as strategic therapeutic targets given their crucial role in DNA repair[21–24]. Our results show that CHK1 inhibition (with the highly CHK1-selective small molecule inhibitor, GDC-0575) extensively enhances the effect of the most efficient DNA-damaging agent used in the clinic for the cure of AML (AraC). This chemo-potentiation strategy not only is beneficial for the increased clearance of AML, but it has implications for adding on to treatment paradigms for elderly patients who cannot tolerate high chemotherapy doses. Accordingly, in our xenograft models, we used a 7-day treatment regimen of low-dose AraC (LDAC; 10 mg/kg), to mirror exposures reported in the clinic[33] and combined it with intermittent dosing of the CHK1i, to marked benefit. While our results broadly confirm and extend conclusions reached by other groups using AML cell lines[21, 23, 24, 34], we have used patient-derived samples and orthotopic engraftment, permitting the assessment of treatment impact in a model system with greater clinical relevance.

Our results are consistent with the first phase 1 clinical report using the combination AraC+CHK1i and showing the combination to be beneficial and relatively well tolerated[35].

In this study, our in vivo data suggest that the addition of CHK1i has an enhancing effect only on AML patient samples that are responsive to AraC treatment as a single agent. In fact, we showed that the AML5 sample is quite resistant to AraC and the addition of CHK1i has little additional effect. We suspect that intrinsic chemoresistance is more likely due to defective cytarabine transport and/or metabolism in this patient, as suggested by other published work[36, 37]. As this patient sample responded better when G-CSF was added to AraC plus CHK1i, this indicates that at least part of the resistance was due to the cell cycle status of the cells.

Interestingly, AraC has been suggested to be mutagenic in most AML patients[25]. In our in vivo models, we observed the generation of AraC-related de novo mutations only in one patient by whole-exome sequencing. The difference in mutation burden observed here might be due to differences in dosage of the drug used or duration of treatment. Remarkably, the addition of CHK1i to AraC treatment in our model prevented the survival of mutagenic clones and thus the generation of new possible driving mutations. The CHK1i itself did not exhibit mutagenic potential in any of the three models tested. This finding highlights a perhaps under-appreciated aspect of chemo-potentiation. By reducing the threshold of DNA damage required for cell killing, checkpoint abrogation strategies may reduce therapy-related mutational load, thereby lowering the probability of acquired resistance and/or generation of secondary cancers. Further analysis will be needed to confirm our preliminary data.

Importantly, one of the main issues when testing new drugs is their target specificity/off-target liabilities, which determines the toxicity profile for normal tissues and hematopoietic cells. In this study, we show that intermittent GDC-0575 exposure, alone or in combination with AraC, has little impact on normal long-term hematopoiesis. Notably, we obtained the same results using both cord blood-derived and adult bone marrow-derived hematopoietic cells, which are known to exhibit different cell cycle features, among other properties. These data are encouraging to clinical development plans, in that if mouse and human pharmacokinetic profiles are similar, the AraC+CHK1i combination promises to eradicate AML disease, while maintaining the potential for a good bone marrow recovery.

Notably, our data with primary AML cells in vitro show that the addition of CHK1i to AraC treatment is important for enhancing the clearance of the bulk AML cells, but has little effect on AraC-mediated LIC reduction. Moreover, cell cycle analysis clearly showed enrichment in G0-phase cells after A and A+C treatment, which suggests that human cells escaping the treatment are quiescent and not exclusively enriched in LICs, in accordance to what we previously described[38]. In support of these results, the addition of CHK1i to AraC on primary AML models in vivo, confirmed the increased reduction in leukemic burden but did not enhance the LIC's depletion as demonstrated by the similar reduced engraftment capacity between A and A+C in secondary transplanted mice. This set of data also suggests that LICs are not more quiescent than leukemic blasts but enter and exit the cell cycle according to environmental signals.

In contrast, A and A+C treatment of normal hematopoietic cells selects for quiescent HSCs, as supported by the higher engraftment in secondary mice. These data, therefore, shed light on a different balance between stemness vs. quiescence between LICs and HSCs. It is indeed largely accepted that LICs, like HSCs, are quiescent and their quiescence has become a crucial therapeutic target[39–41]. Here we show that, differently from HSCs, quiescence is not a characteristic sufficient to select for LICs.

Ultimately, in this study we also improved, in vivo, the efficacy of the A+C treatment by mobilizing, via G-CSF treatment, the residual quiescent AML cell pool. The efficacy of G-CSF in improving chemotherapy has been controversial. One study suggests that it significantly enhances the elimination of human primary AML stem cells in vivo[32] but in another work, no evidence of an increase in complete response rate in the presence of G-CSF was observed[42]. Our data show that in our system the addition of the mobilizer to the A+C therapy caused an almost complete clearance of AML cells, including LICs, which lost their engrafting capacity in secondary mice. Finally, the triple treatment did not negatively impact normal long-term hematopoiesis as it failed to affect the ability of HSCs to engraft in secondary mice. We have therefore demonstrated a safe way to override

---

**Fig. 5** G-CSF enhances the cytotoxicity of AraC+CHK1i on AML in vivo. **a** Treatment regimen for AraC, CHK1i, and G-CSF with primary AML samples and normal hematopoietic cells, in vivo. **b** Percentage of human cells in the bone marrow of mice transplanted with four primary AML samples 1 week post treatment. The treatment started at HE (gray area). Each dot represents an individual mouse. AML5 UT $n = 5$; A+C $n = 4$; A+C+G $n = 3$; AML6 UT $n = 5$; A +C $n = 6$; A+C+G $n = 5$; AML7 UT $n = 4$; A+C $n = 4$; A+C+G $n = 4$; AML8 UT $n = 3$; A+C $n = 4$; A+C+G $n = 3$. Number of total mice = 50. **c** Percentage of human cells in the bone marrow of AML7-transplanted mice 1 week post treatment. **d** Remaining leukemic cells in the bone marrow of AML7-transplanted mice 1 week post treatment shown by hematoxilin & eosin (H&E, left panel) and immunofluorescence (right panel). Green cells are human CD45[+], red cells are Ki67[+], and nuclei are labelled blue. Scale bar is 100 micron for all images. Lower graph represents the ratio of cycling Ki67[+]CD45[+] cells. **e** L-LTC-IC assay on hCD45[+] cells pooled from AML7-injected mice 1 week post treatment. Cells were kept in vitro for 2 weeks in methylcellulose in a LDA assay. Linear regression with 95% confidence bands, ($p$ value = $9.6^{-11}$). A+C $n = 3$; A+C+G $n = 3$. **f** Secondary transplantation of hCD45[+] cells, pooled from AML8-injected mice 1 week post treatment. Each dot represents an individual mouse (A+C $n = 3$; A+C+G $n = 3$). **g** Percentage of human cells in the bone marrow of hBM-transplanted mice 1 week and eight weeks post treatment (UT $n = 4$; G $n = 4$). *$p < 0.05$ and ns = non-significant

concertedly both DNA repair and cell quiescence-mediated AraC resistance in AML.

In summary, this study conducted in multiple clinically relevant primary AML xenograft models, highlights the potential for significant clinical utility of LDAC/CHK1i regimens via reduction of leukemic burden, potential elimination of de novo AraC-mutated cells and enhanced patient survival.

Furthermore, the use of G-CSF to recruit quiescent AML into cell cycle may accelerate the kinetics to achieve "minimal residual disease" status, a triple combination therapy strategy that spares normal long-term hematopoietic stem/progenitor cells (HSPCs) and thereby improves patient recovery. The potential for GDC-0575 as a safe and effective CHK1 inhibitor warrants further investigation in LDAC/G-CSF combinations in the clinic.

## Methods

**Drug treatment.** CHK1 inhibitor (GDC-0575) was kindly provided by Genentech. For in vivo experiments, aliquots of GDC-0575 (10 mg/mL) were stored at −20 °C and diluted in 100 mM sodium citrate buffer immediately prior to each experiment. GDC-0575 was used at 1.8 mg/ml for female mice (~25 g) and 2.6 mg/ml for male mice (~35 g) via oral gavage (final concentration 7.5 mg/kg). For in vitro experiments, aliquots of GDC-0575 (100 μM) were stored at −20 °C and used at a final concentration of 100 nM. AraC (Cytosine β-D-arabinofuranoside C1768, Sigma) was used at 10 mg/kg (commonly used in clinical practice) for in vivo and at 100 nM and 500 nM for in vitro experiments. For in vivo experiments, toxicity of GDC-0575 was assessed in non-engrafted NSG mice using a range of concentrations of GDC-0575 in combination with AraC. 7.5 mg/kg GDC-0575 was the highest concentration to have no significant or lasting adverse effects in mice. Similarly, for in vitro experiments, 100 nM GDC-0575 in combination with 500 nM AraC was the highest concentration non-cytotoxic to MS5 stoma cell. ATR inhibitor (AZ 20, Tocris, Bristol, UK) was used at a final concentration of 0.5 μM.

**RNA silencing.** For silencing of Chk1 expression, two lentiviral vectors were purchased from VectroBuilder, one shRNA against CHK1 (5′-GCAA-CAGTATTTCGGTATAAT-3′) and one shRNA against LacZ used as a control (5′-GTCGGCTTACGGCGGTGATTT-3′). Lentivirus was produced from these constructs by transient calcium phosphate-mediated transfection of 293T cells with the lentiviral backbone plasmid, a vsv-g encoding plasmid and a gag/pol plasmid. Virus supernatants were concentrated by ultracentrifugation to achieve titers of $1 \times 10^9$ IU/ml.

**AML cell lines and primary cell co-cultures.** HL60, KG-1, ML1, U937 AML cell lines and MS-5 feeder cells were originally obtained through ATCC and maintained by the Francis Crick Institute Cell Bank. Each cell line was validated by short tandem repeat (STR) profiling using the PowerPlex 16HS system. AML cell lines were cultured in RPMI 1640 containing 10% heat-inactivated FBS, 2% penicillin, and streptomycin at 37 °C in 5% $CO_2$/95% air. MS-5 feeder cells were cultured in IMDM containing 10% heat-inactivated FBS, 2% penicillin, and streptomycin at 37 °C in 5% $CO_2$/95% air. MS-5 cells were subcultured when reaching 80% confluency. An antibody against mouse Sca-1 (BD Pharmingen, Oxford Science Park, UK) was used to identify the feeder. Primary AML cells were grown in Myelocult H5100 medium (StemCell Technologies, Vancouver) supplemented with 20 ng/ml IL-3, 20 ng/ml G-CSF and 20 ng/ml TPO (PeproTech, London, UK). For co-culture experiments, 2 days before initiating the co-culture, feeder cells were plated onto type-I collagen-coated 96-well or 6-well plates (Falcon) and allowed to reach confluence. One day before starting co-culture, they were irradiated at 6.8 Gy and the culture media exchanged. On day 0 of the co-culture, AML cells were plated at $2 \times 10^5$ cells/ml using the correspondent AML medium. Cells were cultured at 37 °C in 5% $CO_2$-humidified incubators at indicated oxygen concentrations. For short-term culture (STC), cells were kept for 1 week in hypoxia (5% $O_2$) with the indicated drugs: 500 nM AraC and/or 100 nM GDC-0575 (Genentech). For long-term culture/limiting dilution assay (LTC/LDA), human cells from STC were magnetically sorted using hCD45 via the StemSep system (Stem Cell Technologies) according to the manufacturer's instructions and plated on a freshly prepared irradiated MS-5 layer at the indicated concentrations. They were cultured in normoxic conditions with no addition of drug for 4 weeks and half medium changes were performed once a week without disrupting the established feeder layer.

**Primary AML, cord blood and bone marrow cells.** AML samples were obtained after informed consent at St Bartholomew's Hospital (London, UK). The cells were collected and frozen at diagnosis. Details of patient samples are provided in Supplementary Table 1. AML mononuclear cells (MNCs) were isolated by centrifugation using Ficoll-Paque (GE Healthcare Life Sciences, Buckinghamshire, UK). Before using AML MNCs, we performed an immuno-magnetic T-cell depletion by the Easysep T-cell enrichment cocktail (StemCell Technologies,

Vancouver, Canada). Umbilical cord blood (UCB) samples were obtained from normal full-term deliveries after signed informed consent. Both AML and UCB samples collection were approved by the East London Research Ethical Committee (REC: 06/Q0604/110) and in accordance with the Declaration of Helsinki. MNCs were purified by Ficoll-Paque density centrifugation (GE Healthcare Life Sciences, Buckinghamshire, UK) and ammonium chloride red cell lysis. Density-separated cord blood mononuclear cells were magnetically sorted for CD34 via the StemSep system (Stem Cell Technologies) according to the manufacturer's instructions to generate $CD34^+$ cells. Adult bone marrow (BM) $CD34^+$ cells were purchased directly from Lonza (Lonza, Basel, Switzerland).

**Establishment of AML models in immunodeficient mice.** AML cell line ($2 \times 10^6$ cells/mouse), primary AML samples ($10^5$ to $2 \times 10^6$ cells/mouse), hCB $CD34^+$ ($1 \times 10^5$/mouse), and hBM $CD34^+$ ($3 \times 10^5$ cells/mouse) were transplanted into 8–12 weeks old female or male NOD-SCID IL2Rγnull (NSG) mice (The Jackson laboratory, Bar Harbor, ME, USA) using intravenous injection. Twenty-four hours before transplantation, mice were sublethally irradiated at the dose of 3.75 Gy. To assess the level of engraftment, BM samples were aspirated from a long bone while mice were under isoflurane anesthesia, at different time points after transplantation. After the establishment of engraftment (8–12 weeks), mice were randomized into different treatment groups based on the level of engraftment of each mouse so that each group of mice have the same mean level of human $CD45^+CD33^+$ cells. The investigator was thus not blinded to the group allocation. Mice were treated with drugs according to the experimental design. For secondary transplant, mouse BMs of the same experimental condition were pooled and human $CD45^+$ cells were sorted. $0.5–1 \times 10^6$ cells were transplanted into secondary sublethally irradiated NSG mice. All animal work was done under the project license (PPL 70/8904) approved by the UK Home Office and following the CRUK guidelines.

**Flow cytometry and cell sorting.** For analysis and sorting of AML and HSPCs derived from hCB or adult BM, cells were stained with hCD45-PeCy7(Clone: H30, cat: 560915, dilution 1 in 25), mCD45-PerCPCy5.5 (Clone:30-F11, cat: 550994, dilution 1 in 400), Lineage-FITC (lin1, cat: 340546, 1 in 25), CD34-PE (Clone 581, cat: 560941, dilution: 1 in 25), and CD38-APC (Clone HIT2, cat: 555462; dilution: 1 in 25) (BD Biosciences, Oxford, UK). Human grafts in mice were assessed using CD19-FITC (Clone: H1B19; cat: 555412, dilution: 1 in 25), CD33-PE (Cat: WM53, cat: 555450, dilution 1 in 25), CD3-APC (Clone: UCHT1, cat: 561811, dilution: 1 in 25), hCD45-PeCy7 (Clone: H30, cat: 560915, dilution 1 in 25), and mCD45-PerCPCy5.5 (Clone:30-F11, cat: 550994, dilution 1 in 400) (BD Biosciences). Luciferase-transduced HL60 and U937 cells were identified and sorted based on their GFP expression. Non-viable cells were excluded by DAPI staining. Appropriate isotype-matched antibodies were used as controls. Flow cytometry analysis was performed using an LSRII flow cytometer (BD Biosciences). Cell sorting was performed using a FACS Aria or INFLUX (BD Biosciences).

**Bioluminescence imaging.** Isofluorane-anesthetized animals were imaged using the Xenogen IVIS imaging system 5–10 min after D-luciferin (Caliper Life Sciences, Cambridge, UK) was injected intraperitoneally (150 mg/kg). Bioluminescence images were taken from both ventral and dorsal sides of the mice. The photons emitted from HL60-Luc and U937-Luc, expressed as Flux (photons/s/cm²/steradian), were quantified and analyzed using the "Living image" software (Caliper Life Science).

**Cell proliferation by XTT assay.** AML cell lines were seeded at $1 \times 10^4$ cells/well in 96-well plates in triplicate, and subjected to different treatment conditions. After 24 h of incubation, cell proliferation was measured with the XTT Cell Proliferation Kit II (Roche, Mannheim, Germany) in accordance with the manufacturer's protocol. The spectrophotometric absorbance of each well was determined using a SPECTRA max PLUS 384 (Molecular Devices, Sunnyvale, CA). The wavelength for measuring absorbance was 450 nm and the reference wavelength was 650 nm.

**Apoptosis assay.** AML cell lines were seeded at $2 \times 10^5$ cells/well in 24-well plates in triplicate. After 24 h of incubation with various drugs, Annexin V labeling was used to quantify the effects of treatment on apoptosis. Thirty microliters of 10× Annexin V binding buffer (BD Pharmingen) were added to 270 μl of the resuspended cells and mixed. Then, 2 μl of directly conjugated Annexin V Alexa Fluor 647 (Molecular Probes) was added to the cells before incubation at 37 °C for 15 min. DAPI was added to the cells before analysis on a BD LSR-2 flow cytometer.

**Cell cycle.** Intracellular immuno-staining for Ki67 was used to determine the cell cycle status. AML cells from co-cultures were first incubated with anti-human CD45-PeCy7 (Clone: H30, cat: 560915, dilution 1 in 25) and Sca-PE (Clone: D7, Cat: 553108, dilution 1 in 400) antibodies (BD Biosciences, Oxford, UK), washed with PBS, fixed in 1 ml of PBS with 2% methanol-free formaldehyde at room temperature (RT) for 10 min and washed twice with PBS. Cells were then permeabilized with 1 ml of PBS containing 0.1% Triton-X-100 (TX; Sigma, Dorset, UK) for 10 min at RT. After washing, cells were incubated with Alexa Fluor 647-Ki67 antibody (eBioscience, San Diego, USA) at 4 °C for 1 h. Cells were

resuspended with PBS 2% FBS buffer containing DAPI (2 mg/ml) and analyzed by FACS.

**Immunofluorescence**. AML cell lines were allowed to attach to poly-L-lysine-coated slides for 30 min. They were then fixed with 4% paraformaldehyde at RT for 10 min, permeabilized with PSB/0.1% TX for 10 min, blocked with 10% serum (donkey or goat) for 30 min at RT, incubated with primary antibodies at 4 °C overnight (O/N) and then with appropriate secondary antibodies for 1 h at RT. Slides were mounted with fluorescence mounting medium (Dako, Cambridgeshire, UK) containing DAPI for nuclear staining. Sections were then visualized using a fluorescent microscope (Zeiss AxioVision2; Zeiss, Welwyn Garden City, UK). Images were acquired using Axiocam MRC with Axiovision 4.7 software. Antibodies used: $\gamma$H2Ax (05-636, Millipore), CHK1Ser[345] (133D3, Cell Signaling Technology).

**Immunohistochemistry**. Spinal column samples were harvested, formalin-fixed, and decalcified with 17% EDTA (Osteosoft, Millipore) during 7 days. Then samples were processed, paraffin embedded, and transverse sectioned (5 $\mu$m) for histological studies. Hematoxylin/eosin staining, immunohistochemestry (IHC), and immunofluorescence (IF) studies were performed by standard methods. For IHC and IF, heat-induced antigen retrieval was performed when required. Primary unconjugated antibodies employed were anti-Ki67 (Abcam, Ab16667), and anti-humanCD45 (Dako, M0701). Secondary fluorescent antibodies were from Invitrogen and secondary biotinylated antibodies were from Jackson. After IF, slices were treated with Sudan black 0.1% to reduce auto-fluorescence in bone marrow tissue.

**Western blot analysis**. Total protein extracts (20 $\mu$g) were run on a denaturing 10% SDS-polyacrylamide electrophoresis (PAGE) gel and transferred by wet transfer to nitrocellulose membranes. The primary antibodies used were CHK1 (sc-22799), CDC25A (sc-7389) from Santa Cruz Biotechnology (Santa Cruz, CA, USA), CHK1 Ser345 (133D3), CHK1 Ser296 (90178), pTyr15 CDK2 (9111), and GAPDH (2118) from Cell Signaling Technology, Danvers, MA, USA), CDK2 (010145) from BioScience and pCDC25A (156574) from Abcam. Protein bands were visualized using an enhanced chemiluminescence visualization system (ECL Plus, Amersham Life Sciences).

**Chemotherapy treatment of in vivo models**. NSG mice were injected intravenously with $1 \times 10^5$–$10^6$ cells of AML and $1$–$3 \times 10^5$ cells of hCB CD34$^+$/hBM CD34$^+$. After demonstrating AML engraftment at 9–11 weeks through FACS analysis of tibia bone marrow aspiration, mice were treated accordingly to proper 7-day treatment regimen with daily 10 mg/kg AraC via subcutaneous injection, 7.5 mg/kg CHK1i suspension administered via oral gavage on every other day schedule, and/or 300 $\mu$g/kg G-CSF (Chugai Pharma UK) administered daily for 5 days via intraperitoneal injection. One week after the final dosing, mice were killed by cervical dislocation. The femurs, tibias, and pelvis were dissected and flushed with PBS. Red blood cells were lyzed via ammonium chloride. Cells were stained with human-specific FITC-conjugated anti-CD19, PE-conjugated anti-CD33, PE-Cy7-conjugated anti-CD45, and PERCP-conjugated anti-murine CD45 antibodies. Dead cells and debris were excluded via DAPI staining. A BD LSR II flow cytometer was used for analysis. Flow cytometry analysis was performed with FlowJo software (Tree Star, Oten, Switzerland). More than 100,000 DAPI-negative events were collected. Engraftment of AML was said to be present if a single population of mCD45$^-$hCD45$^+$CD33$^+$CD19$^-$ cells was present without accompanying mCD45$^-$hCD45$^+$CD33$^-$CD19$^+$cells.

**Statistical analysis**. Statistical analyses were performed using GraphPad Prism Version 6.0 f (GraphPad Software). Error bars indicate the SEM of data from replicate experiments. The significance of difference between samples within figures was confirmed using multivariable one-way ANOVA unpaired analysis corrected for multiple comparisons. Observed differences were regarded as statistically significant if the calculated $p$ value was below 0.05. $*p < 0.05$; $**p < 0.001$; $***p < 0.0001$; $****p < 0.00001$. For xenotransplantation analysis, a non-parametric Mann–Whitney unpaired test was applied.

**Whole-genome amplification**. Whole-genome amplification (WGA) of cells from mice was performed using illustra Single Cell GenomiPhi DNA Amplification Kit (GE Healthcare Life Sciences). WGA amplification was performed according to the manufacturer's instructions along with a negative control with no DNA and a reaction with human tissue genomic DNA as a positive control.

**Whole-exome sequencing and data analysis**. Three AML patients were selected for whole-exome sequencing (WES), using DNA from day 0 CD33$^+$ cells (primary bone marrow sample) in all cases and DNA from CD3$^+$ T cells as a paired constitutional DNA source. In addition, DNA from CD45$^+$CD33$^+$ cells derived from mice following xenotransplantation were also subjected to WES analysis. WES was only performed in mice where enough xenografted cells were recovered (three patient samples transplanted into 34 mice). qDNA (non-WGA, 100 ng) or WGA

DNA (500 ng) was processed for exome library capture (Agilent SureSelect V5) and sequenced on the Illumina HiSeq2500 (Paired end V4 chemistry) according to the manufacturer's instructions. Initially, base calling was performed by the Illumina RTA software (Supplementary Fig. 6). Demultiplexing and conversion of basecalls to fastq files was performed by Casava version 1.8.2. Data alignment, realignment, and recalibration were performed using Burrows–Wheelers aligner (BWA-MEM v0.7.12)[43] and GATK[44, 45], respectively; according to Broad Institute best practices. VarScan (v2.4.1[46, 47] was subsequently used to perform somatic variant calling on mpileup files generated by Samtools[48] for paired tumor and normal (CD3$^+$ T cells) tissue as well as for xenografted samples. Following on, data were then passed through ANNOVAR[49] using refseq annotation and variants not found in dbSNP138 and 1000 genomes databases at >0.01 population allele frequency were passed. Genomic duplicated regions were also filtered out unless associated with known mutations. Somatic mutations were subsequently passed when somatic $p$ value (VarScan, Fisher's exact $t$-test) was <0.01, had ≥7 supporting reads, were present in paired normal tissue at less than 20% of the paired tumor tissue and had a allele burden of >5%.

Stringent secondary analysis was performed to obtain high confidence single-nucleotide variants (mutations). For xenografted CD45$^+$CD33$^+$ cells, candidate mutations were only scored if they were present in more than one mouse within the same treatment category, and/or found concurrently in Day 0 sample or any of the untreated mice for the same patient. Variants with homopolymer runs of five or more flanking the mutation were also filtered out. Variant that are present in genes which are frequently mutated in myeloid malignancies were included in the final list even if they failed any of the above-mentioned data filtering criteria.

**Exome coverage**. Exome coverage was calculated utilizing GATK DepthOf-Coverage against intervals that the Agilent V5 Sureselect kit capture probes were based on. Mean base coverage across the targeted region ranged from 10–281 X and for >90% of the exome base coverage was >10 reads in all cases (apart from one WGA t-cells) (Supplementary Table 4).

**Data availability**. The majority of data generated or analyzed during this study are included in this published article (and its supplementary information files). Others are available from the corresponding author on reasonable request. The NGS data generated in this study are deposited in the European Nucleotide Archive (ENA) under the accessible code (Study: PRJEB23171- project: enba-STUDY-Francis Crick Institute-20-10-2017-16:34:55:352-226).

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

## Acknowledgments

We are indebted to patients who gave samples. We also gratefully acknowledge William Grey for critical reading of the manuscript. We thank the Biological Resource Unit (BRU), the Flow Cytometry, In Vivo Imaging, Light Microscopy, Advanced Sequencing, Computational Biology, and Experimental Histopathology core facilities at the Francis Crick Institute. We thank Mrs Shohreh Beski (Obstetric Department, Royal London Hospital, London, UK) and her team as well as Mrs Jashu Patel for helping in the collection of umbilical cord blood. This work was supported by the Francis Crick Institute, which receives its core funding from Cancer Research UK (FC0010045), the UK Medical Research Council (FC0010045), and the Wellcome Trust (FC001045), and by Genentech (research grant to D.B.).

## Author contributions

A.D.T. performed the research, analyzed data, and wrote the manuscript. K.R.P., A.B. and W.G. performed some research and analyzed the data. S.M. performed genome amplification and prepared the samples for whole-genome sequencing as well as analyzed the results. A.G. analyzed WGS data. J.G. provided AML samples and relevant patients' information; A.S. provided the sequencing analysis. E.B. helps in the design of the experiments, in the analysis and interpretation of the data. D.B. directed the research, analyzed data, and wrote the manuscript. All authors approved the manuscript.

## Additional information

**Competing interests:** E.B. is an employee of Genentech. D.B.'s group was supported by a research grant from Genentech. The remaining authors declare no competing financial interests.

