## [Peer Review File · Nature Communications]

Reviewers' comments:

Reviewer #1 (Remarks to the Author):

In this manuscript, Di Tulio et al. report that the orally bioavailable Chk1 inhibitor GDC-0575 enhances anti-AML effects *ex vivo* and *in vitro*, that cells persisting after treatment with the combination are often quiescent, and that addition of G-CSF to the cytarabine/Chk1 inhibitor further enhances the anti-leukemic effect without significantly enhancing cytarabine-induced mutagenesis in the AML cells or toxicity to normal marrow progenitors and precursors. These results are discussed in the context of potential future treatment of AML in the clinic.

The present manuscript represents a tremendous amount of work and is interesting, with a number of novel aspects as outlined by the authors in the Discussion. However, the manuscript could potentially be improved by attention to the following items:

1. One of the hallmarks of research is the ability of others to repeat it. Accordingly, this reviewer is surprised that the authors would attempt to publish (and Nature Communications would agree to review) a manuscript in which the structure of a key novel reagent is not shown or previously disclosed.

2. The strategy that is the major focus of the first portion of the paper (cytarabine + Chk1 inhibitor) has been previously tested in the clinic. While the results appeared promising in the phase I trial (J. Karp et al., *Clin. Cancer Res.* 18:6723, 2012), a subsequent randomized phase 2 trial failed to show any benefit of adding Chk1 inhibitor to cytarabine in relapsed AML (J. A. Webster, ASH abstract 2563, 2015). Accordingly, the authors might want to at least mention this prior clinical work and speculate why their results are so much more promising.

3. It is unclear how the cytarabine concentration of 500 nM was chosen for *in vitro* and *ex vivo* studies. This is 5- to 10-fold higher than the cytarabine concentration sustained with conventional cytarabine treatment (e.g., 100 mg/m²/d x 7 days) and would hardly be considered analogous to low-dose cytarabine as suggested in the Discussion. The choice of this high cytarabine concentration for *in vitro/ex vivo* work needs to be justified and discussed.

4. There are potential problems with the manner in which the apoptosis assay was performed. As currently stated, a final concentration of 1X annexin binding buffer (presumably 150 mM NaCl/1 mM CaCl₂) was added to cells already in isotonic medium. If the protocol described in this manuscript is followed, cells are in roughly 300 mM NaCl when annexin V binding is detected. One wonders what sort of artifact (positive or negative) the high tonicity of the assay introduces and whether it is the same for all groups.

5. It is unclear from the methods whether p values are appropriately corrected for multiple comparisons.

6. The designation of synergy is essentially based on a fractional multiplication method (legend to Table S2). This fractional multiplication method has been criticized by statisticians (M.C. Berenbaum. *Pharm. Rev.* 41:93, 1989). One wonders whether synergy is present when more conventional analysis such as isobologram analysis is performed.

Reviewer #2 (Remarks to the Author):

Di Tullio et al. provide a straightforward and clear study utilizing a CHK1 inhibitor GDC-0575 in combination with AraC and G-CSF. Although the concept of combining these inhibitors is well established, their study tests the combination of these drugs using both AML cell lines and primary

AML patient samples with detailed assessment of LIC activity and mutational status. The authors also provide evidence that the combination of therapies provide a therapeutic window and spare the normal hematopoietic compartment. This is an important study for the development of this drug into the clinic but several issues should be addressed.

Major points:

1. If the authors want to claim and title their manuscript that the triple combination is the best therapeutic strategy they should then provide additional data to support their conclusions including adding the arm of AraC + G-CSF with these patients. Is there a benefit because of the increased proliferation or mobilization out of the niche? The change in the quiescent pool in Figure 5 should also be better documented or quantified. Some evidence for their mechanism for why the triple combination is superior should be provided.

2. In Fig2D Cell Cycle analysis it is not clear what is statistically significant. For example, the authors claim that for AML patient #5 the weak response is due to an increase in quiescent cells, which provides rationale for G-CSF used in Fig. All of the AMLs tested are already in G0 so this reasoning isn't fully supported.

3. Did the authors follow any other AML patient in leukemia burden and survival (as shown in Fig3D)? Can the authors include the survival curve of AML9 complementing Fig S2A-C? Furthermore, it would be interesting to know if AML6 is also showing differences in spleen weights as shown for AML9?

4. The point that AraC + CHK1 blocking mutations outgrowth should be tempered as it was only observed in one patient.

.

Minor points:

1. In Figure 1S is A+C statistically significantly increasing apoptosis in the cell lines compared to A or C alone?

2. Is AML8 or AML9 shown in Fig. 3B on the right panel as another patient is used in the other panel.

Reviewers' comments:

Reviewer #1 (Remarks to the Author):

In this manuscript, Di Tulio et al. report that the orally bioavailable Chk1 inhibitor GDC-0575 enhances anti-AML effects *ex vivo* and *in vitro*, that cells persisting after treatment with the combination are often quiescent, and that addition of G-CSF to the cytarabine/Chk1 inhibitor further enhances the anti-leukemic effect without significantly enhancing cytarabine-induced mutagenesis in the AML cells or toxicity to normal marrow progenitors and precursors. These results are discussed in the context of potential future treatment of AML in the clinic.

The present manuscript represents a tremendous amount of work and is interesting, with a number of novel aspects as outlined by the authors in the Discussion. However, the manuscript could potentially be improved by attention to the following items:

1. One of the hallmarks of research is the ability of others to repeat it. Accordingly, this reviewer is surprised that the authors would attempt to publish (and Nature Communications would agree to review) a manuscript in which the structure of a key novel reagent is not shown or previously disclosed.

As mentioned in the paper, a new manuscript describing the structure of the GDC-0575 is being prepared by Genentech (Elizabeth Blackwood, et al. In Preparation) and thus we could not provide more structural details in the present paper. Nevertheless, in addition with the specificity data already presented (see kinome Sup. Table 1), we provide below a comparison between GDC-0575 and two structurally related Chk.1 inh from Genentech GNE-900 (see publication: Blackwood E et al., Mol Cancer Ther, 2013; 12 (10): 1968-80) and GDC-0425 (which has just been reported in a phase 1 clinical trial: see Infante JR et al., Clinical Can Res. 2017:23 (10): 2423-2432).

Legend: Effect of different chk.1 inh on the AML cell lines survival treated with Ara-C.

2. The strategy that is the major focus of the first portion of the paper (cytarabine + Chk1 inhibitor) has been previously tested in the clinic. While the results appeared promising in the phase I trial (J. Karp et al., Clin. Cancer Res. 18:6723, 2012), a subsequent randomized phase 2 trial failed to show any benefit of adding Chk1 inhibitor to cytarabine in relapsed AML (J. A. Webster, ASH abstract 2563, 2015). Accordingly, the authors might want to at least mention this prior clinical work and speculate why their results are so much more promising.

There is indeed at present only one phase 1 clinical trial using Ara.C plus Chk.1 inh that has been completed which was mentioned by the reviewer. In the conclusion of the paper reporting the results, the authors mentioned that further phase II trials are needed.

In addition, even in the reported randomized phase 2 trials reported at ASH by Webster et al., the authors despite reporting similar response rates between Ara and Ara+Chk1.inh, show improvement in responses in the combination treatment in patients in first relapse and following BMT. Thus, it is clearly too early to speculate potential differences between our preclinical data and preliminary clinical reports so far.

To our knowledge, as exemplified by the last ASH meeting, a number of pharmaceutical companies are still focusing on the use of chk.1inh and Ara.C in haematological malignancies. In fact, a new clinical trial using GDC0425 and Ara.C in AML has also started.

In addition, here we provide pre-clinical evidence that the triple combination of Ara.C +Chk.1inh plus G-CSF will be even more efficient than Ara-C plus chk.1 inh.

3. It is unclear how the cytarabine concentration of 500 nM was chosen for in vitro and ex vivo studies. This is 5- to 10-fold higher than the cytarabine concentration sustained with conventional cytarabine treatment (e.g., 100 mg/m²/d x 7 days) and would hardly be considered analogous to low-dose cytarabine as suggested in the Discussion. The choice of this high cytarabine concentration for in vitro/ex vivo work needs to be justified and discussed.

We would like to thank the reviewer for his/her comments. We originally tested a lower dose of Ara.C 100 nM ± Chk.1 inh on four different cell lines (see below). We nevertheless decided to use a

higher dose of Ara.C *ex vivo* as we wanted to mostly test the efficacy of adding chk.1 inh on the more Ara.C resistant cells. We show clearly that even at this dose, a good percentage of cells (both cell lines and primary cells) are still able to survive. Nevertheless, as shown in figure 2B, the addition of chk.1inh significantly decreases the number of residual live cells.

Nevertheless, when we tested the Leukemic long-term initiating (L-LTC-IC) frequency of these residual cells, we could clearly see that chk.1 inh did not have any effect on L-LTC-IC depletion.

In vivo, we wanted to be in a more relevant clinical setting and thus we use a low dose Ara.C (10mg/kg) and originally tested different doses of chk.1 inh. We show (see figure below), that chk.1inh at 7.5mg/kg have similar activity than chk.1 inh at 10 mg/kg and thus used Ara.C 10mg/kg and chk.1 inh at 7.5 mg/kg for all the experiments.

4. There are potential problems with the manner in which the apoptosis assay was performed. As currently stated, a final concentration of 1X annexin binding buffer (presumably 150 mM NaCl/1 mM CaCl₂) was added to cells already in isotonic medium. If the protocol described in this manuscript is followed, cells are in roughly 300 mM NaCl when annexin V binding is detected. One wonders what sort of artifact (positive or negative) the high tonicity of the assay introduces and whether it is the same for all groups.

We performed the Annexin staining following the general staining protocol provided by BD. All cells were treated the same way.

5. It is unclear from the methods whether p values are appropriately corrected for multiple comparisons.

In most of our analysis, we focus our comparison on Ara-C versus Ara + chk.1 inh and thus did not use multi-parameters comparison for this part. We used Log-rank test for our sickness curves.

6. The designation of synergy is essentially based on a fractional multiplication method (legend to Table S2). This fractional multiplication method has been criticized by statisticians (M.C. Berenbaum. Pharm. Rev. 41:93, 1989). One wonders whether synergy is present when more conventional analysis such as isobologram analysis is performed.

We could not use the isobologram analysis as chk.1 at the dose tested as no effect on the percentage of apoptosis and after discussion with our biostatistician, we decided to use the fractional multiplication method instead. We were not aware of the potential criticized /controversy concerning this test and thus have eliminate the term “synergy” in the study.

Reviewer #2 (Remarks to the Author):

Di Tullio et al. provide a straightforward and clear study utilizing a CHK1 inhibitor GDC-0575 in combination with AraC and G-CSF. Although the concept of combining these inhibitors is well established, their study tests the combination of these drugs using both AML cell lines and primary AML patient samples with detailed assessment of LIC activity and mutational status. The authors also provide evidence that the combination of therapies provide a therapeutic window and spare the normal hematopoietic compartment. This is an important study for the development of this drug into the clinic but several issues should be addressed.

Major points:

1. If the authors want to claim and title their manuscript that the triple combination is the best therapeutic strategy they should then provide additional data to support their conclusions including adding the arm of AraC + G-CSF with these patients.

Based on our data showing that the residual leukemic blast present in the mice after Ara.C plus Chk.1 inh were indeed quiescent, we tested whether the addition of G-CSF will be effective to further decrease the leukemic burden and test whether these residual cells were or not resistant to the combination Ara+chk.1 inh.

We mentioned indeed that using a similar xenotransplantation model, Saito Y et al. (*Nature biotechnology* 28, 275-280 (2010). elegantly show that G-CSF when co-administrate with Ara.C decrease the leukemic stem cells pool *in vivo*. The goal was thus not to repeat their experiment directly but test whether the addition of G-CSF to Ara.C plus chk.1 inh will indeed be superior than Ara.C plus chk.1 only. We show that it was the case. In Figure 5 of the paper, we show that even in patient AML8 where still a few leukemic cells were present after the triple combination therapy, no secondary leukemic engraftment was detected providing evidence that the LSC pool was eradicated.

Once again it is the first paper demonstrating using functional assay *in vivo* the efficacy of the triple combination (Ara.C + Chk1inh + G-CSF). Nevertheless we agree with this reviewer that because we did not test the combination of Ara.C plus G-CSF in our study we did not formally prove that the triple combination is superior. We nevertheless believe it is the case as we show that Ara.C plus chk.1inh show superiority to Ara.C alone and that G-CSF treatment should only increase the percentage of cells being sensitive to Ara.C or Ara.C+chk.1inh.

Is there a benefit because of the increased proliferation or mobilization out of the niche?

We do indeed believe that the addition of G-CSF not only help mobilized leukemic stem cells out of their niche but also increase their proliferative potential as shown in Figure S5. Thus, Ara-C plus chk.1inh. are thus able to kill a larger proportion of leukemic stem cells.

The change in the quiescent pool in Figure 5 should also better documented or quantified. Some evidence for their mechanism for why the triple combination is superior should be provided.

Figure 5D provide evidence via immunostaining that persistent leukemic cells after Ara.C plus chk.1 inh plus G-CSF are indeed not cycling. We have now added the quantification of the ratio of human CD45+ that are Ki67+ between A+C versus A+C+G.

Furthermore, the mechanism by which the triple combination is superior is provided in Figure S5. Indeed, we show that G-CSF treatment is able to not only mobilized leukemic cells in the blood but also increase the proliferation of the cells (via increase in ki67+ cells being detected). Thus, we believe it is the combination of both mobilization out of the niche and the increase in proliferation that increase the number of cells being susceptible to Ara.C + Chk.1 inh treatment.

2. In Fig2D Cell Cycle analysis it is not clear what is statistically significant. For example, the authors claim that for AML patient #5 the weak response is due to an increase in quiescent cells, which provides rationale for G-CSF used in Fig. All of the AMLs tested are already in G0 so this reasoning isn't fully supported.

The statistically significance concerned the increase in G0 cells comparing A vs A+C and UT. In all samples tested there is a significant increase in G0 after A or A+C compare to UT except for AML 5 which is not responding. Our rational to use G-CSF was the fact that residual cells after A+C inh were enriched in G0. This was not really the case for patient 5 which is a non-responder. We added the stat in Fig 2D.

3. Did the authors follow any other AML patient in leukemia burden and survival (as shown in Fig3D)?

We indeed added another sickness curve from AML-9 samples in Figure S2 showing similar benefit of the addition of Chk.1inh to Ara.C alone.

Can the authors include the survival curve of AML9 complementing Fig S2A-C? Furthermore, it would be interesting to know if AML6 is also showing differences in spleen weights as shown for AML9?

We unfortunately did not weight the spleen of mice transplanted from other samples.

4. The point that AraC + CHK1 blocking mutations outgrowth should be tempered as it was only observed in one patient.

We have now indeed temper our conclusion of this analysis as we only observed the increase in mutations in the Ara.C group for one patient.

Minor points:

1. In Figure 1A is A+C statistically significantly increasing apoptosis in the cell lines compared to A or C alone?

Indeed, the apoptosis is significantly increase in A+C in Fig 1B compare to A or C alone. We have now corrected this in the revised version.

2. Is AML8 or AML9 shown in Fig. 3B on the right panel as another patient is used in the other panel.

Different patients were used in different panel for high and medium engraftment. Thus, the identity of the AML used in each panel is correct.

Reviewers' comments:

Reviewer #1 (Remarks to the Author):

In this revised manuscript, Di Tulio et al. report that the orally bioavailable Chk1 inhibitor GDC-0575 enhances anti-AML effects of cytarabine and that G-CSF treatment further enhances the effects on AML cells without long-term effects on normal hematopoietic progenitors. While the results remain interesting, the major concerns listed in the prior review have not been satisfactorily addressed, markedly diminishing enthusiasm for publication in its present form. The major concerns are:

1. Lack of reproducibility—without the structure of GDC-0575, other investigators in the scientific community will not be able to assess reproducibility of the present results. This is a major limitation that impedes scientific progress. Manuscripts that are “in preparation” sometimes get published and sometimes do not. In view of the current emphasis on scientific rigor and reproducibility, it behooves the authors to assure that others have the opportunity to reproduce these results. This reviewer can see three potential strategies to provide this assurance: a) publish the structure in this manuscript, b) withdraw the present manuscript and resubmit once the structure has been published, or c) publish an explicit statement, certified by Genentech, indicating that GDC-0575 will be made available to interested investigators who wish to reproduce the results without conditions (e.g., without MTAs) and without delay to facilitate reproduction of the results.

2. Failure to cite important prior work in this field—the fact that AraC/Chk1 inhibitor combinations have previously been tested clinically (e.g., Karp et al., *Clinical Cancer Research* 18:6723, 2012) still is not acknowledged or cited in this manuscript.

3. The amount of cytarabine used throughout the manuscript remains extremely high and completely unjustified. First, the concentration of 500 nM used in vitro remains 5- to 10-fold higher than the clinically sustainable concentration, e.g., the concentration achieved with 200 mg/m² x 7 days continuous infusion in the clinic (see ref. 33 for the statement that this is standard of care). The use of a 5- to 10-fold higher concentration than is achieved with this standard regimen still is not adequately discussed in the present manuscript. Second the dose administered to mice in the present manuscript, 10 mg/kg AraC, is still described as low-dose AraC (p. 14). Moreover, the manuscript cites reference 33 as showing that 10 mg/kg AraC is “commonly used in clinical practice” (p. 18). Review of reference 33 indicates that LDAC is defined as 20 mg total dose per day in humans, which (for the average 70 kg human adult) comes out to 0.28 mg/kg/day, or 1/30th of the dose administered in the present study. How the authors decided that 10 mg/kg corresponds to “low dose” AraC or “commonly used in clinical practice” remains unclear but must be stated and justified in the manuscript.

One possibility (which this reviewer prefers) would be for all in vitro and ex vivo experiments to be repeated at 50-100 nM AraC, which is the clinically relevant concentration based on the pharmacokinetic work of Plunkett and coworkers. At the very least, the figure showing that similar, albeit less impressive, results were obtained with 100 nM in cell lines, which is currently cited in the text as “data not shown,” could be included as a supplemental figure.

4. This reviewer remains concerned that the annexin V binding assay has been performed improperly. According to the BD Biosciences website (<http://www.bdbiosciences.com/us/applications/research/apoptosis/buffers-and-ancillary-products/annexin-v-binding-buffer-10x-concentration/p/556454>), 10X annexin V binding buffer contains 1.4 M NaCl. Adding 270 ul of cells (in isotonic buffer) to 30 ul of 10X annexin V binding buffer as described in the Methods (pp. 21-22) will result in a final NaCl concentration of roughly 280 nM NaCl, which is nonphysiologic. The authors' response to this point does nothing to address this issue.

5. This reviewer is extremely concerned about the lack of rigor in the statistical analysis. A large number of experiments contain multiple comparisons (including but not limited to Figs. 1A, 1B, 1C, 2B, 2C, 2D, etc.). All of the experiments should be examined to determine whether corrections for multiple comparisons need to be made. The authors are encouraged to consult a statistician on this point. Without this correction, the vast majority of p values in this manuscript deliberately overstate the statistical significance of the findings, potentially leading to incorrect conclusions.

6. This reviewer previously questioned whether synergy had been demonstrated. Although the authors acknowledge in their rebuttal that they have not rigorously demonstrated synergy, the manuscript still states (e.g., p. 4) that synergy has been demonstrated.

7. Upon review of this manuscript a second time, this reviewer is concerned that additional claims within the manuscript also are not rigorously supported by the data. Examples include:

a. The claim that cytarabine activates Chk1 (p. 5 and Discussion)—in support of this, the authors point to Chk1 Ser345 phosphorylation. To in fact show that Chk1 has been activated, one needs to show that a Chk1 substrate has been phosphorylated. The way to do this is to blot for the Chk1 autophosphorylation event (e.g. Ser296) or another phosphorylated substrate (e.g., phospho-CDC25A). This has not been done.

b. The claim that CDC25A has been activated (p. 5)—in support of this, the authors report that CDC25A expression is increased (Fig. S1A). Again, the way one shows activation of a phosphatase is to show that phosphorylation of its substrate (e.g., CDK2) has been decreased, not by showing increase of the phosphatase itself. Moreover, according to current understanding, activation of Chk1 should lead to downregulation of CDC25A, not upregulation. If current understanding is correct, either Chk1 has not been activated (see point a) or the CDC25A staining, which has not been shown in this manuscript to be specific in any way, is spurious. At the very least, blots in Fig. S1C should be probed for CDC25A. If it is increased, then some attempt should be made to reconcile this finding with the claim that Chk1 is activated.

c. GDC-0575 is said to inhibit Chk1 in the present study. I see no evidence anywhere in this manuscript that Chk1 has been inhibited in AML cells at concentrations that have been applied. At the very least, one would expect to see that phosphorylation of one or more Chk1 substrates has been diminished by the treatment of AML cell lines and AML cells with GDC-0575. If this could be accompanied by experiments showing that GDC-0575 also inhibits phosphorylation of the same substrates in xenograft experiments (especially the human xenograft experiments), that would be even better.

8. The entire manuscript needs to be proofread carefully. Many of the modifications made in response to previous comments are grammatically incorrect. Moreover, many journal titles appear incorrect as well. For example, "Journal of clinical oncology: official journal of the American Society of Clinical Oncology" is not a journal title that appears in PubMed or on the ASCO website.

Reviewer #2 (Remarks to the Author):

The authors have adequately responded to the critiques.

Reviewers' comments:

Reviewer #1 (Remarks to the Author):

In this revised manuscript, Di Tulio et al. report that the orally bioavailable Chk1 inhibitor GDC-0575 enhances anti-AML effects of cytarabine and that G-CSF treatment further enhances the effects on AML cells without long-term effects on normal hematopoietic progenitors. While the results remain interesting, the major concerns listed in the prior review have not been satisfactorily addressed, markedly diminishing enthusiasm for publication in its present form. The major concerns are:

1. Lack of reproducibility—without the structure of GDC-0575, other investigators in the scientific community will not be able to assess reproducibility of the present results. This is a major limitation that impedes scientific progress. Manuscripts that are “in preparation” sometimes get published and sometimes do not. In view of the current emphasis on scientific rigor and reproducibility, it behooves the authors to assure that others have the opportunity to reproduce these results. This reviewer can see three potential strategies to provide this assurance: a) publish the structure in this manuscript, b) withdraw the present manuscript and resubmit once the structure has been published, or c) publish an explicit statement, certified by Genentech, indicating that GDC-0575 will be made available to interested investigators who wish to reproduce the results without conditions (e.g., without MTAs) and without delay to facilitate reproduction of the results.

We are happy to report that after discussion with Genentech, they agree to let us include the structure of the inhibitor which is now included in Supplementary Figure S1.

2. Failure to cite important prior work in this field—the fact that AraC/Chk1 inhibitor combinations have previously been tested clinically (e.g., Karp et al., *Clinical Cancer Research* 18:6723, 2012) still is not acknowledged or cited in this manuscript.

We have now added this reference in the text.

3. The amount of cytarabine used throughout the manuscript remains extremely high and completely unjustified. First, the concentration of 500 nM used in vitro remains 5- to 10-fold higher than the clinically sustainable concentration, e.g., the concentration achieved with 200 mg/m² x 7 days continuous infusion in the clinic (see ref. 33 for the statement that this is standard of care). The use of a 5- to 10-fold higher concentration than is achieved with this standard regimen still is not adequately discussed in the present manuscript. Second the dose administered to mice in the present manuscript, 10 mg/kg AraC, is still described as low-dose AraC (p. 14). Moreover, the manuscript cites reference 33 as showing that 10 mg/kg AraC is “commonly used in clinical practice” (p. 18). Review of reference 33 indicates that LDAC is defined as 20 mg total dose per day in humans, which (for the average 70 kg human adult) comes out to 0.28 mg/kg/day, or 1/30th of the dose administered in the present study. How the authors decided that 10 mg/kg corresponds to “low dose” AraC or “commonly used in clinical practice” remains unclear but must be stated and justified in the manuscript.

One possibility (which this reviewer prefers) would be for all in vitro and ex vivo experiments to be repeated at 50-100 nM AraC, which is the clinically relevant concentration based on the pharmacokinetic work of Plunkett and coworkers. At the very least, the figure showing that similar, albeit less impressive,

results were obtained with 100 nM in cell lines, which is currently cited in the text as “data not shown,” could be included as a supplemental figure.

The dose used *ex vivo* of 500nM has mentioned already has been chosen as it is a safe dose for the MS.5 feeder cells and we could see an efficient effect of Ara.C on cell lines and patients' samples used. We agree that this dose is quite high but we do not agree with the reviewer that we could really compare an *ex vivo* dose to what patients received *in vivo*. Indeed, we use this dose once and thus difficult to compare this to a continuous infusion of 200mg/m² for 7 consecutive days which is as mentioned by the reviewer the usual dose used in combination with 3 days pre-injection of dauxorubicin. Nevertheless, we agree with the reviewer that the use of lower dose of Ara.C 50-100nM might allow us to show an even improved effect of the addition of chk.1 inh. Thus, based on the additional data we already provided for this reviewer in the last revision, we have now incorporated this data in the supplemental figure S3.

Concerning the *in vivo* dose use of 10 mg/kg AraC being similar to the low dose Ara.C, we believe it is. We gave a 0.2 mg /day /mouse. The assumption that you could calculate the dose/rate equivalent between mouse and human simply by corrected this to the weight is not correct. A mouse of around 20g has a much large surface area (roughly 0.006 m²) compare to a human of 70Kg (surface area of 1.9m²). We thus used the guidance for mouse-human dosing which take into account the difference in body surface area (see reference: <https://www.ncbi.nlm.nih.gov/pmc/articles/PMC4804402/>).

The low dose Ara.C in human is 40mg per day (dose splits in two administrations) This corresponds to a 70kg man to: 40/70 = 0.57mg/kg. Using the conversion from table 2 (see website above), 12.3 x 0.57= 7mg/kg for mice based on their metabolism/surface area. For a 28 g mouse this is 0.2mg per day.

4. This reviewer remains concerned that the annexin V binding assay has been performed improperly.

According to the BD Biosciences website

(<http://www.bdbiosciences.com/us/applications/research/apoptosis/buffers-and-ancillary-products/annexin-v-binding-buffer-10x-concentrate/p/556454>), 10X annexin V binding buffer contains

1.4 M NaCl. Adding 270 ul of cells (in isotonic buffer) to 30 ul of 10X annexin V binding buffer as described in the Methods (pp. 21-22) will result in a final NaCl concentration of roughly 280 nM NaCl, which is nonphysiologic. The authors' response to this point does nothing to address this issue.

We agree with the reviewer that in the protocol it is said that the 10 X buffer should be diluted in water for a 1x solution before used. Nevertheless, we also diluted the 10X buffer to a 1 x solution by using only 30ul in total of 270 ul of cells suspended in PBS. We used this protocol for both control and treated cells and thus our comparative analysis is still valid. Furthermore, we have used this similar protocol before (see Griessinger et al., Stem Cell Trans Med, 2014;3(4):520-9).

Nevertheless, to confirm that the protocol used here did not influence the results obtained, we compared the two protocols on a HL.60 cell line ± Ara.C side by side (see graph below).

Apoptosis on HL.60

5. This reviewer is extremely concerned about the lack of rigor in the statistical analysis. A large number of experiments contain multiple comparisons (including but not limited to Figs. 1A, 1B, 1C, 2B, 2C, 2D, etc.). All of the experiments should be examined to determine whether corrections for multiple comparisons need to be made. The authors are encouraged to consult a statistician on this point. Without this correction, the vast majority of p values in this manuscript deliberately overstate the statistical significance of the findings, potentially leading to incorrect conclusions.

We took advise to our statistician and have recalculate all our *ex vivo* analysis using one way Anova for all multiple analysis. We used a non-parametric test for all *in vivo* analysis. The statistical paragraph in the method has been changed.

6. This reviewer previously questioned whether synergy had been demonstrated. Although the authors acknowledge in their rebuttal that they have not rigorously demonstrated synergy, the manuscript still states (e.g., p. 4) that synergy has been demonstrated.

We agreed with this reviewer that without proper testing for synergy we should not use this term in the text. We are grateful that this reviewer pointed out that in page 4, a reference about synergy was kept. We have now remove this statement.

7. Upon review of this manuscript a second time, this reviewer is concerned that additional claims within the manuscript also are not rigorously supported by the data. Examples include:

a. The claim that cytarabine activates Chk1 (p. 5 and Discussion)—in support of this, the authors point to Chk1 Ser345 phosphorylation. To in fact show that Chk1 has been activated, one needs to show that a Chk1 substrate has been phosphorylated. The way to do this is to blot for the Chk1 autophosphorylation event (e.g. Ser296) or another phosphorylated substrate (e.g., phospho-CDC25A). This has not been done.

We have adding a WB on Chk.1 Ser 296 to show activation of the pathway after Ara.C. We added these results in the Western Blot of Fig S2C.

b. The claim that CDC25A has been activated (p. 5)—in support of this, the authors report that CDC25A expression is increased (Fig. S1A). Again, the way one shows activation of a phosphatase is to show that phosphorylation of its substrate (e.g., CDK2) has been decreased, not by showing increase of the phosphatase itself. Moreover, according to current understanding, activation of Chk1 should lead to downregulation of CDC25A, not upregulation. If current understanding is correct, either Chk1 has not been activated (see point a) or the CDC25A staining, which has not been shown in this manuscript to be specific in any way, is spurious. At the very least, blots in Fig. S1C should be probed for CDC25A. If it is increased, then some attempt should be made to reconcile this finding with the claim that Chk1 is activated.

c. GDC-0575 is said to inhibit Chk1 in the present study. I see no evidence anywhere in this manuscript that Chk1 has been inhibited in AML cells at concentrations that have been applied. At the very least, one would expect to see that phosphorylation of one or more Chk1 substrates has been diminished by the treatment of AML cell lines and AML cells with GDC-0575. If this could be accompanied by experiments showing that GDC-0575 also inhibits phosphorylation of the same substrates in xenograft experiments (especially the human xenograft experiments), that would be even better.

To answer both comments made (b and c) at once, we added a new WB done on HL.60 ± Ara.C ± chk.1 inh. Looking at the protein expression of CDC25A, we could not detect any significant changes in either A or Chk.1 inh or Ara+Chk.1 inh. This is probably due to the fact that de/stabilization of the CDC25 family of phosphatases is generally challenging to see in all cell lines. We nevertheless show that the pCDC25A was

increase after Ara.C and phosphorylation of CDC25A has been associated with inactivation. We nevertheless agree with the reviewer that the most informative way to show changes in CDC25A activity, is to show that CDC25A substrate was changing. We thus look at pCDK2 and could clearly show an increase in pCDK2 after Ara.C and a decrease after chk.1 inh. These results have been integrated into Figure S.2D. As we now added this new Blot, we agree with this reviewer that the IF of CDC25A in Supplemental Fig S2A is not informative and thus deleted the panel.

8. The entire manuscript needs to be proofread carefully. Many of the modifications made in response to previous comments are grammatically incorrect. Moreover, many journal titles appear incorrect as well. For example, "Journal of clinical oncology: official journal of the American Society of Clinical Oncology" is not a journal title that appears in PubMed or on the ASCO website.

We have now proof read the revised version and reformatted the references.

REVIEWERS' COMMENTS:

Reviewer #1 (Remarks to the Author):

In this revised manuscript, Di Tullio et al. report the effects of cytarabine +/- Chk1 inhibitor +/- G-CSF on AML cell lines and clinical samples. They continue to report that Chk1 inhibitor enhances killing by cytarabine and G-CSF reduces dormancy-related resistance.

The manuscript in its present form addresses most of my previous concerns. Two concerns remain, and a series of minor items still need attention.

1. The statistical analysis as described in the rebuttal and in the Methods remains problematic. ANOVA is a test designed to analyze the differences between means of a series of groups, i.e., to tell whether one of a series of groups differs from the others. Reporting the p value from the ANOVA, which is what the Methods now indicate, is meaningless. What is usually reported is the result of pairwise comparisons. My original question is whether these pairwise comparisons (performed after ANOVA indicated a difference between groups) had been appropriately corrected for multiple comparisons. It appears now from the rebuttal and Methods that the results of the ANOVA rather than pairwise comparisons have been reported, yet the figures still indicate pairwise comparison of groups (i.e., p values between different "bars" on the figure). For the values to be meaningful, the p values should be generated by ANOVA with post-hoc comparisons; and you must indicate whether the resulting p values from the pairwise comparisons have been appropriately corrected for multiple comparisons. The Methods should reflect this and indicate the method by which the corrections were done. I suggest consulting a statistician if there remains a question.

2. It appears that the results provided in response to query #4 (the use of an incorrect method for annexin V binding) need a bit more attention. First, the fact that the authors published with it previously does not indicate the correctness of the method (or the soundness of the previous conclusions derived with it). Second, examination of the dot plots provided indicates that the number of living cells (left lower quadrant) after cytarabine treatment is substantially lower when the assay is performed in PBS vs. (as we now agree is correct) water. In addition, the total number of annexin V positive cells (early apoptosis, right lower quadrant + late apoptosis, right upper quadrant) goes from 26% to 34% for the same sample under different conditions, i.e., a relative increase of 30%. Given these differences, I suggest that the authors include the two figures related to their nonstandard assay conditions as supplemental figures to convince readers that the apoptosis assays are valid. I think this is preferable to continuing to perpetuate the nonstandard method and leaving readers wondering whether apoptosis has been accurately measured or not.

There are also a small number of minor issues that require attention:

3. In Fig. 1G and some of the other figures it looks as if the results with cytarabine alone are significantly different from untreated. These differences between groups should also be analyzed rather than "cherry picking" the differences to analyze.

4. Many of the bar graphs (main figures AND supplement) show a different fill from one treatment to another. If there is significance to the changes in fill, there should be a "key" with each graph to indicate what these changes in fill denote. If there is no significance to these changes in fill and their pattern, why are they included in the figures?

5. According to the legend, Fig. 3D shows a "sickness curve." The Y axis indicates the percentage of "healthy" animals. What were the criteria for healthy vs. sick? Did this include blood counts, microCT, or some other objective measure? If not, i.e., if the well-being of the animals was simply monitored by inspection, which is very subjective, wouldn't survival have been a more objective

endpoint?

6. Line 119 indicating that previous reports have described enhanced killing by DNA damaging agents in the presence of Chk1 inhibitors—based on the titles, it appears that references 20, 21 and 24 might also have established this point. If so, they should be cited in this sentence.

7. Line 145—it is unclear what is meant by “adding chk.1” to chemotherapy. Is this another typo that has been missed?

8. Despite the indication that references have been checked and corrected, there remain problems with ref. 21 (a nonexistent journal title that ends in “official journal of..”) as well as refs. 6 and 7 (strange spacing in journal titles). Once again, this reviewer suggests careful proof reading before resubmission.

REVIEWERS' COMMENTS:

Reviewer #1 (Remarks to the Author):

In this revised manuscript, Di Tullio et al. report the effects of cytarabine +/- Chk1 inhibitor +/- G-CSF on AML cell lines and clinical samples. They continue to report that Chk1 inhibitor enhances killing by cytarabine and G-CSF reduces dormancy-related resistance.

The manuscript in its present form addresses most of my previous concerns. Two concerns remain, and a series of minor items still need attention.

1. The statistical analysis as described in the rebuttal and in the Methods remains problematic. ANOVA is a test designed to analyze the differences between means of a series of groups, i.e., to tell whether one of a series of groups differs from the others. Reporting the p value from the ANOVA, which is what the Methods now indicate, is meaningless. What is usually reported is the result of pairwise comparisons. My original question is whether these pairwise comparisons (performed after ANOVA indicated a difference between groups) had been appropriately corrected for multiple comparisons. It appears now from the rebuttal and Methods that the results of the ANOVA rather than pairwise comparisons have been reported, yet the figures still indicate pairwise comparison of groups (i.e., p values between different "bars" on the figure). For the values to be meaningful, the p values should be generated by ANOVA with post-hoc comparisons; and you must indicate whether the resulting p values from the pairwise comparisons have been appropriately corrected for multiple comparisons. The Methods should reflect this and indicate the method by which the corrections were done. I suggest consulting a statistician if there remains a question.

We would like to apologize for not making this clearer in our last version of the paper and our response to the reviewer. We have indeed already performed a one way Anova corrected for multiple comparisons and are reported all pairwise comparisons for all our ex vivo results and use a non-parametric Mann-Whitney unpaired test for our in vivo analysis.

We nevertheless agree that in the last version, we did not incorporate in the graph all the multiple comparisons statistic obtained and only provide the C and A+C pairwise comparison results. We thought the graph will have been too confusing if added all comparison and statistic. Nevertheless, we agree that we should incorporate all the comparisons obtained and thus have find a way to add all the statistical analysis into each graph.

2. It appears that the results provided in response to query #4 (the use of an incorrect method for annexin V binding) need a bit more attention. First, the fact that the authors published with it previously does not indicate the correctness of the method (or the soundness of the previous conclusions derived with it). Second, examination of the dot plots provided indicates that the number of living cells (left lower quadrant) after cytarabine treatment is substantially lower when the assay is performed in PBS vs. (as we now agree is correct) water. In addition, the total number of annexin V positive cells (early apoptosis, right lower quadrant + late apoptosis, right upper quadrant)

goes from 26% to 34% for the same sample under different conditions, i.e., a relative increase of 30%. Given these differences, I suggest that the authors include the two figures related to their nonstandard assay conditions as supplemental figures to convince readers that the apoptosis assays are valid. I think this is preferable to continuing to perpetuate the nonstandard method and leaving readers wondering whether apoptosis has been accurately measured or not.

We agree that we used a slightly different method of dilution of annexin than what is usually recommended. Nevertheless, we show in the last response to this reviewer that there was no difference in the % of Annexin positive between the two methods. The reviewer mentioned that there was an increase of 30% in apoptotic cells (26 to 34% which in reality is only 26 % to 29% from the FACS plot provided to this reviewer. This plot is one only out of the four repeats done. Thus, when plotting everything we show no differences in apoptotic cells. We are now including all the analysis of this same experiment done in quadruplicate and show that there is no statistical significant difference in apoptotic or pre-apoptotic cells between the two group (H2O treated versus PBS). Adding this result in the supplementary data is thus not justified. As we are happy to comply with Nat Communication guideline for transparency in peer review, our response and the data provided to all the reviewers during this process will all be available to everybody under a supplementary peer review file.

Effect of Ara.C 500nM on HL.60 viability

There are also a small number of minor issues that require attention:

3. In Fig. 1G and some of the other figures it looks as if the results with cytarabine alone are significantly different from untreated. These differences between groups should also be analyzed rather than “cherry picking” the differences to analyze.

As mentioned in our answer to the first point, we have now incorporated all multiple comparisons and statistical significance in the Figures.

4. Many of the bar graphs (main figures AND supplement) show a different fill from one treatment to another. If there is significance to the changes in fill, there should be a “key” with each graph to indicate what these changes in fill denote. If there is no significance to these changes in fill and their pattern, why are they included in the figures?

We tried to use a different symbol when using primary AML versus cell lines and normal HSPCs. We also wanted to differentiate, treatment from CHK.1 inh compare to CHK.1 KD by using different filing of the bar graph. In Fig S7 we also had to incorporate the G-CSF treatment. We are sorry if that might add some confusion but we believe that the nomenclature of each treatment group in each figure is clearly annotated.

5. According to the legend, Fig. 3D shows a “sickness curve.” The Y axis indicates the percentage of “healthy” animals. What were the criteria for healthy vs. sick? Did this include blood counts, microCT, or some other objective measure? If not, i.e., if the well-being of the animals was simply monitored by inspection, which is very subjective, wouldn’t survival have been a more objective endpoint?

We indeed represent in Fig 3D the sickness curve and compare healthy versus sick as we had to follow the UK Home Office guidelines. We are not allowed to wait for the mice to die. We had to kill mice as soon as a weight loss of 20% has been reached. We incorporate this information in the Figure legend.

6. Line 119 indicating that previous reports have described enhanced killing by DNA damaging agents in the presence of Chk1 inhibitors—based on the titles, it appears that references 20, 21 and 24 might also have established this point. If so, they should be cited in this sentence.

We have now incorporated the references mentioned by the reviewer.

7. Line 145—it is unclear what is meant by “adding chk.1” to chemotherapy. Is this another typo that has been missed?

Based on Nat Communicosn policy, we had to shorten this subheading and thus now is: “CHK1i increases the effect of Ara.C on AML in vivo”.

8. Despite the indication that references have been checked and corrected, there remain problems with ref. 21 (a nonexistent journal title that ends in “official journal of...”) as well as refs. 6 and 7 (strange spacing in journal titles). Once again, this reviewer suggests careful proof reading before resubmission.

We have now manually corrected the references. The error came from an endnote problem.